# Teachers' perceptions of the differential impacts of a universal, school-based social and emotional learning intervention: A thematic framework analysis

Suzanne Hamilton[1]*, Jan R. Boehnke[2], Neil Humphrey[1], Pamela Qualter[1]

1 Manchester Institute of Education, The University of Manchester, United Kingdom, 2 School of Health Sciences, University of Dundee, Dundee, United Kingdom

* suzanne.hamilton@manchester.ac.uk

## Abstract

Trials of universal school-based (USB) social and emotional learning (SEL) interventions have reported that individual and socio-demographic characteristics may moderate outcomes, but it is not clear how. Teachers are key stakeholders in SEL and are involved in the implementation of interventions, so they can offer insights into patterns of responsiveness and intervention impact. This qualitative study aimed to explore teachers' perceptions of the differential impacts of the PATHS curriculum, a USB SEL intervention. Data were generated in semi-structured interviews with 105 implementing teachers as part of a trial of PATHS in 23 primary schools across Greater Manchester, and analysed using thematic framework analysis. Six main themes and 22 subthemes formed the final thematic framework. Teachers suggested that students demographic backgrounds influenced both engagement with and responsiveness to the intervention and that a certain initial level of social and emotional skills were needed as foundation on which to build further learning. The potential of teachers' expectations and beliefs about pupils' characteristics to impact on implementation quality and judgements of impact are discussed. Implications for practice and future research are considered.

## Introduction

### Universal school-based social and emotional learning interventions

Social and emotional learning (SEL) refers to the process of acquiring and effectively applying the knowledge, attitudes, and skills necessary to understand and manage emotions, set and achieve positive goals, feel and show empathy for others, establish and maintain positive relationships, and make responsible decisions. Universal school-based SEL (USB SEL) interventions are delivered in schools to all pupils regardless of existent skills, difficulties, or risk. That means they are non-stigmatising

**Data availability statement:** The data are stored on secure encrypted servers held by the University of Manchester. We are unable to make data available for anyone outside of the research team at the University of Manchester. This is because the data were collected prior to 2014, and consent was not obtained from participants or from University Research Ethics Committee for the public sharing of their data. Whilst the data are anonymised, they contain potentially identifiable and sensitive information relating to participants and their pupils. NH was a principal investigator of the main PATHS evaluation; we were granted secure remote access to this data by Dr Alexandra Hennessey (trial manager). All data requests for data from the PATHS evaluation, in which this data were collected, should be submitted to Professor Neil Humphrey for consideration. Access to available anonymised data may be granted. Access to data may also be granted through researcher collaboration with the University of Manchester. External researchers should contact the Office for Open Research (openresearch@manchester. ac.uk).

**Funding:** This study forms part of SH's PhD research, which is funded by the Kavli Trust (https://kavlifondet.no/en/). Grant no. Kavli2021-0000000019, awarded to NH and PQ. The PATHS trial, in which data were collected, was funded by the National Institute for Health Research (https://fundingawards.nihr.ac.uk/award/10/3006/01) . Grant no.10/3006/01, awarded to NH. Neither funders play any role in study design, data collection, analysis, or reporting.

**Competing interests:** The authors have no competing interests to declare.

and have the potential to prevent multiple problems that are predicted by shared risk factors, or that may develop later in life, but have not yet emerged [1,2]. Meta-analytic evidence has shown that the main effects of universal school-based SEL interventions include enhancements to pupils' social and emotional skills [3–7], improved attitudes and prosocial behaviours, decreased conduct problems, and reduced emotional distress [5,7]. Effects are strongest in the first 12 months post-intervention, but some improved outcomes, such as future social relationships, have been maintained up to 18 years later [8].

Within the field of SEL, the focus is still predominantly on main effects [1]. This aligns with the public health approach of reducing a smaller risk in the wider population, to shift the risk profile of the whole population [9]. However, it can be expected that individuals will respond differently and even in unique ways to the same intervention, and that not all population subgroups will benefit from in an intervention in the same way or to the same extent [10]. It is important to understand how the effectiveness of a treatment varies across a universal population to allow young people from all backgrounds to experience similar benefits from an intervention.

## Examination of differential effects

To date, there has been limited examination of differential effects in USB SEL trials, and studies that have investigated such effects are lacking in consensus. Some have identified compensatory effects, whereby those with weaker competencies at baseline make greater gains than those with stronger skills [11], for example, a trial of the *Second Step®* SEL intervention reported significant improvements in social–emotional competence and behaviour for children who started the school year with skill deficits relative to their peers [12]. In contrast, some studies have reported 'Matthew' (also known as 'rich get richer') effects, whereby those who have stronger skills at baseline gain absolutely and relatively more than others from the intervention [13]; in a trial of the Promoting Alternative THinking Strategies (PATHS) curriculum, children with low initial levels of problem behaviour made significantly greater gains than those with high initial levels of problem behaviour [14]. Others have reported 'Goldilocks effects', whereby an intervention is most effective for those who have neither too many nor too few baseline difficulties [15]; a review of USB SEL interventions for older children, ages 12–18 years of age, found that interventions aimed at preventing depression, anxiety, and violence are more effective for young people who have elevated, but subclinical symptoms [4]. This hypothesis suggests that some level of existing social and emotional competency is needed in order to benefit from a USB SEL intervention. Concerningly, recent studies have reported harmful effects for particular subgroups [16].

## The role of pupils' individual and socio-demographic characteristics

Trials of USB SEL interventions have reported that certain individual and socio-demographic characteristics can moderate outcomes, perhaps due to differential experiences in the development of social and emotional skills throughout childhood.

Gender differences in emotion socialisation [17–19], emotion regulation [20,21], and emotion expression [22] are believed to contribute to gender differences in social and emotional competency throughout childhood. Data indicate that boys tend to express more emotional regulation difficulties than girls, from preschool [20] to middle childhood [21]. Gender differences in aggressive behaviour are also widely reported, with boys being more aggressive than girls, and typically more physically than indirectly aggressive, whilst girls are typically more indirectly aggressive than physically aggressive [23]. Gender differences in emotion expression have been reported in early and middle childhood, with boys showing more externalising emotions, such as anger, and girls showing more positive emotions, as well as internalising emotions, such as sadness, anxiety, and sympathy [22]. In adolescence, this pattern changes, and girls have been found more likely than boys to report emotion regulation difficulties [21] and externalising emotions [22].

Studies investigating subskills within the broad construct of emotional regulation have found differences in the emotion regulation competencies of boys and girls and suggest that boys may be more competent in areas related to understanding and regulating emotions, whilst girls may be more proficient in regulating emotion in social situations when interacting with peers, such as sharing, being kind, helpful, and considerate of their peer's feelings [20]. Studies of the strategies used by children and youth to regulate emotions report that girls are more likely to seek social support to regulate emotion and engage in dysfunctional rumination, whilst boys report more passivity, avoidance, and suppression [24]. Furthermore, research also finds that boys and girls are socialised to express emotion differently and that cultural expectations may reward expression of particular emotions and different approaches to emotional coping by gender [25]. Thus, it is important to determine whether all genders can experience similar benefits from USB SEL interventions.

Gender is associated with differential outcomes in some trials: girls have been found to benefit more than boys from interventions targeting internalising symptoms [26–28], whilst boys have been found to benefit more than girls from interventions targeting externalising behaviours [29,30]. However, the picture is far from clear, with one trial reporting boys' externalising behaviours worsened post-intervention [31].

Children with special educational needs or disabilities (SEND) are also more likely to face difficulties and report lower social and emotional competency compared to their peers without SEND [32]. Whilst this is likely due to a complex interaction of different factors, it is held that difficulties with executive functioning and self-regulation play an important role [33,34]. Additionally, difficulties with emotion knowledge [34] and self-concept [35] among those with SEND are associated with increased internalising and externalising problems. Differences in social cognition and social skills have also been identified as contributing factors to mental health difficulties among children with SEND [36,37], and children with SEND experience lower quality peer relationships, fewer friendships [38], and increased bullying [39]. Given this, it may seem reasonable to believe SEL interventions are of particular relevance for Children with SEND, yet a recent review found that only 20 USB SEL intervention studies out of 269 reviewed (7.4%) examined how outcomes differ for students with disabilities. Only 28.3% studies even reported data on student disability status and 4.1% of studies explicitly excluded students on the basis of their disability [40]. In England, over 1.6 million school children have an identified SEN, a substantial number that equates to over 17% of all pupils [41].

Another group who are likely to have increased need for SEL is children from socio-economically disadvantaged households, who are more likely to have social, emotional and/or behavioural difficulties [42,43]. Parental poverty-related stress can disrupt parenting and family relationships, which negatively impacts children's social and emotional development [44–46]. Low socio-economic status has been strongly associated with poorer self-regulation [47], externalising difficulties [48,49], and internalising difficulties [39,50] (NHS Digital, 2022; Vizard et al., 2020) and children from low income families are much more likely to develop mental health difficulties [39,49,51,52].

Free school meal (FSM) eligibility is a common indicator of economic disadvantage [53]. Pupils eligible for FSM underachieve academically, falling significantly behind their wealthier peers [54] and experience higher rates of persistent absence and school exclusions [55]. While family income is a factor, Bourdieu's concept of cultural capital offers further insight [56,57]. Cultural capital is a crucial mechanism through which family SES transmits into children's education [58].

This includes cultural knowledge, language skills, and understanding the 'rules of the game', which are often transmitted by families of higher socioeconomic status. Bourdieu and Passeron argue that the education system favours middle- and upper-class cultural values, disadvantaging working-class children who lack this capital and are often negatively stereotyped by peers and teachers [59]. Given teachers' role in shaping emotions [60], these biases may negatively impact the social and emotional development of disadvantaged students. Teachers' own emotions, reactions, and teaching about emotions are "undergirded by beliefs" and likely send socialisation messages to children, which may be especially impactful for children living at socioeconomic risk [61].

Low-income status and FSM eligibility has been found to moderate outcomes in trials, with children from this group reported to benefit more from some USB SEL interventions [27,62,63]. However, this is not always the case: a trial of the FRIENDS for Life intervention, which aims to reduce anxiety, reported an increase in anxiety and depression in pupils eligible for free school meals (FSM; a proxy measure for economic disadvantage) [64].

Ethnic minority groups are disproportionately represented in low-income households [65,66] and therefore more likely to experience the negative effects of financial hardship and poverty. However, research on the impact of USB SEL interventions among children from ethnic minorities, particularly in the UK, is woefully lacking and hampered by small sample sizes [67–70]. It is clear, though, that experiences of racism have a negative impact on social and emotional development: strong consistent positive relationships have been found between racial discrimination and both internalising and externalising disorders in children and young people [71–75]. Boys from black Caribbean and mixed-race/ethnic backgrounds are over-represented in school exclusions [55] and adolescent girls of mixed-race/ethnicity report lower self-esteem than single race/ethnicity peers from black and minority ethnic backgrounds [76]. Since children's social and emotional development occurs within the context of cultural norms of emotion understanding and emotion expression [77], SEL interventions that prioritise white, middle-class ideals of emotional competency may lack relevancy for minoritised groups [78,79]. It is essential to determine whether USB SEL interventions are effective for culturally minoritised groups.

Overall, evidence on the effectiveness of USB SEL interventions for subgroups is inconclusive because of a lack of reporting on students' sociodemographic characteristics in trials [3,5]. Furthermore, it is likely that an interaction of factors at the individual level, family, peer group and school level will influence the impact of an intervention, and quantitative methods that consider each characteristic in isolation are too simplistic. It is necessary to consider how the interaction between multiple disadvantages can impact on children's social and emotional development because these interactions can create unique effects [80] that may not be addressed by USB SEL interventions. Combinations of types of disadvantage can have multiplicative effects, for instance, having both SEND and a parent who is less engaged in education is associated with a greater educational penalty than the sum of their individual penalties [81]. Higher SES may not have the same protective benefits for black youth as for white youth, possibly due to structural inequalities, cultural values [82], and increased discrimination faced by black students from more wealthy backgrounds [74]. Speaking English as a second language may prevent children from receiving SEND diagnoses and support because language difficulties may mask core features of the SEND [83]. Qualitative inquiry has great utility to investigate differential impacts through its potential to generate rich insights into the nuances in the mechanisms and social processes that can influence intervention effectiveness.

### The role of implementation quality

Implementation quality refers to *how well* facilitators deliver material and engage and interact with participants during implementation of an intervention [84,85]. It is posited that high-quality delivery also requires generalisation of the content outside of the intervention's scripted lessons [86]. Implementation quality has been identified as a significant moderator of USB SEL intervention outcomes [5]: 'better' implemented programs typically yield significantly better outcomes than the 'less well' implemented programs [87].

Whilst there is conceptual ambiguity surrounding implementation quality [88], it is held that implementing teachers' interaction styles with pupils are important elements of delivery, including their affective engagement, sensitivity, and responsiveness to pupils in spontaneous interactions outside of the intervention [89]; implementation quality is sensitive to the individual relationships between teachers and pupils. SEL delivery may require a different style of teaching to other curriculum subjects and include clinical process skills, such as reflective listening and promoting a safe and supportive environment [84,90]. Within psychotherapy, the therapeutic relationship between therapist and client has been identified as a central component to change and one of the most important predictors of therapeutic outcomes [91]. Within child and adolescent mental health services, therapists' behaviours and interpersonal style have been shown to have a significant impact both on the therapist-client relationship and on therapeutic outcomes [92]. Therapist behaviours such as warmth, supportiveness, expectations, concern for and validation of children's feelings have positive impacts on engagement in therapy, whilst behaviours such as pushing the child to talk, forgetting something the child had previously said, or not acknowledging their emotions, negatively impacts the therapeutic relationship and outcomes [92]. Similarly, a study of implementation of a school-based mindfulness SEL program, teachers reported that building trusting relationships with students through respect, personal regard, and integrity was essential in promoting their engagement with the program [93].

Yet, research in SEL is centred on how an intervention can improve children's skills, with little attention to the role of adults [94]. Research examining the influence of teacher factors on implementation quality and intervention effectiveness has focused on teachers' beliefs and attitudes towards SEL in general, as well as specific programs, their self-efficacy in implementing a program [95,96], and their perceptions of school support [97] (Domitrovich et al., 2019). There has also been some attention to emotional exhaustion, burnout, and personality characteristics [89]. There are likely other teacher characteristics that could impact on the implementation of SEL.

## Teacher beliefs and expectations

A wide body of literature demonstrates that teachers' expectations affect students' academic outcomes [98], but how teachers' expectations may impact students' social and emotional outcomes is largely unexplored. This warrants attention, since teachers'

expectations for pupils are influenced by their own class-based values and beliefs [99,100], and these beliefs and values shape the instruction and learning environment in which SEL programs occur [101]; for instance, teachers' beliefs about appropriate behaviour and emotion expression will be shaped by their own cultural values, which, given that the school workforce is predominantly white, female and middle class [102], may differ from many of their students'.

Research focusing on academic outcomes has found that teachers' expectations are influenced by information about pupils' backgrounds [103]: teachers have generally been found to hold lower expectations for pupils from disadvantaged backgrounds [98,104,105] and ethnic minority groups [106], and favour students from ethnic majority groups and more affluent families over those from less affluent families [107]. Studies of teacher's ratings of students' behaviour have reported similar biases: a review of bias in teacher ratings of student behaviour found that students' behaviour is judged more harshly when their culture (in terms of country of origin, region of origin, or socioeconomic status) differed from that of the teacher [108]. In a study exploring the implementation of the USB SEL TOOLBOX intervention among 4–11-year-old children from predominantly low-income backgrounds, teachers underestimated their students' social and emotional competence. Teachers perceived their typical student as having below-average social-emotional competence, but when compared to the school district's normative sample, students' scores on social-emotional measures were actually relative to the norm [109].

Low teacher expectations, communicated through verbal and non-verbal cues, and differential learning opportunities can lead pupils to underestimate their abilities, negatively affecting their self-efficacy, self-concept, and motivation [98]. This can result in a self-fulfilling prophecy of underachievement [98,103] or a self-sustaining prophecy that reinforces

existing behaviours [110]. These negative expectations may also hinder pupils' social and emotional development and damage the teacher-pupil relationship, impacting their receptiveness to social and emotional learning.

### Participant responsiveness

Participant responsiveness refers to the degree to which the program stimulates the interest or holds the attention of participants [111] and may include indicators such as levels of participation and enthusiasm [112]. Participant engagement and responsiveness is closely tied to implementation quality and should be examined when investigating why some groups may benefit more or less from an intervention, since if participants are not engaged, they will not receive the positive benefits of the SEL intervention [87]. Despite their theoretical importance, quality of delivery and participant responsiveness are less frequently studied processes in intervention research [88].

### Qualitative studies of teachers' perceptions

Qualitative methods are best placed to examine quality and responsiveness as they can investigate the "hows and whys" of implementation, and what surrounds and interacts with implementation processes [113]. Teachers lead the implementation of USB SEL interventions, so can offer detailed insights and contextualised understandings of pupils' responsiveness to, and differential benefit from, an intervention. Despite the utility of these methods to explore such nuances, there is a paucity of qualitative research in this area, meaning that the contexts and circumstances in which intervention responsiveness and benefit may vary is overlooked. This study addresses this important gap in the literature.

Where qualitative methods *have* been employed to explore teachers' views of the impacts of USB SEL programs [93,114–123], the focus has been on teacher experiences of implementation, rather than child-level factors influencing implementation. Perceptions of intervention impact are generally discussed in relation to all students as a whole. Only a small number of qualitative studies report teachers' perceptions relating to differential effectiveness [93,116] and SES is the most commonly featured influence [116]. However, even within those studies there is little exploration of the mechanisms through which SES may affect intervention effectiveness. This study aims to address this gap.

### The present study

The present study utilised data generated in the implementation and process evaluation (IPE) strand in a two-year randomised controlled trial of the Promoting Alternative THinking Strategies (PATHS) curriculum [124]. In this trial, PATHS was implemented in Year 3, 4, 5 and 6 classes (children aged 7–9 years at baseline) in 23 primary schools across Greater Manchester throughout 2012–2014 [125]. The trial was approved by the University Research Ethics Committee at the University of Manchester (reference number 11470). Consent from participants was provided in writing.

The aims of the present study were to understand teachers' perceptions of who benefits the most and least from a USB SEL intervention, and why they thought this was the case. Specifically, the study aimed to address the following research questions:

i. How do teachers describe the varied outcomes of the PATHS curriculum on their students, particularly among students from marginalised groups?

ii. How do teachers understand and explain these varied outcomes?

iii. What relationships do teachers describe between contextual factors and students' learning in the PATHS curriculum?

### Theoretical assumptions

The present study was designed from a critical realist epistemological stance, with the position that all thought and action is socially located [126] and that unobservable social structures cause observable events that are partly

accessible to the researcher through interpretation of subjective experiences [127]. The underlying assumption of the study is that teachers can tell us something real about the differential impacts of PATHS in a specific context, and that whilst this knowledge is fallible and imperfect [128], we can "seek out the presence and effects of causal or generative mechanisms" (128, p.33).

**A note on terminology used in this article**

**Gender.** The authors recognise that there is a spectrum of gender identities and expressions defining how individuals identify themselves and express their gender. In this study, gender refers to sex as male or female because schools in England are required to record pupils as either male or female [129] and this is the language used by teachers in the data.

**Socio-economic disadvantage.** Participants used varying language to refer to poverty and socio-economic disadvantage. We have presented participants own words as much as possible. Participants use of the word "disadvantaged" was interpreted to refer to socio-economic disadvantage, i.e., children from low-income households. Within the context of education in England and Wales, the Department for Education (DfE) "considers a child to be *disadvantaged* if they have been registered for FSM at any point in the past six years; are looked after by the local authority; or have left local authority care" ([130], p.14). "Deprivation" and "children from deprived households/ backgrounds/ areas" were interpreted as referring to children from low-income households living in areas of high poverty. Whilst low income is a central component of deprivation, deprivation is conceptualised more broadly as "the lack of socially perceived necessities, typically across the domains of income, employment, education, skills and training, health and disability, crime, barriers to housing and services, and living environment" ([131], p.10).

## Methods

### Participants and context

Participants were 105 teachers of Key Stage 2 pupils (Years 3–6; aged 7–11 years) from the 23 primary schools across Greater Manchester that implemented PATHS in the trial. Recruitment took place between January and April 2012. Verbal and written consent was obtained prior to interviews. Teachers averaged 8 years of classroom experience, were predominantly female (81%), and were educated to postgraduate level (51%). Teachers delivered PATHS to 2294 children in Key Stage 2 (Years 3–6; ages 7–11) in the academic years 2012–2013 and 2013–2014 [125]. Trial schools mirrored those of primary schools in England in respect of attendance, attainment and the proportion of pupils speaking English as an additional language (EAL), but were larger than the national average, had higher proportions of pupils of pupils eligible for FSM, and lower proportions identified as having SEND [125].

Thirty-four per cent of teachers reported having 2–5 years of experience implementing other SEL programs prior to adopting PATHS. In preparation for delivering PATHS, teachers in PATHS schools received "1 initial day and 1 half-day follow-up of training, and were aided by trained external coaches, who offered ongoing technical support and assistance (e.g., lesson modelling, observation and feedback) throughout the school year" ([131], p.2).

### The PATHS curriculum

The Promoting Alternative THinking Strategies (PATHS) curriculum [124] was designed as a targeted intervention for children with hearing impairment. It was later adapted as a universal intervention designed to promote social and emotional competence through the development of skills across five conceptual domains: self-control, emotional understanding, positive self-esteem, relationships, and interpersonal problem-solving skills [132]. PATHS is designed to be taught on a regular basis, at least two times per week in 30-minute sessions, with supplementary activities added at least once a week.

## Data collection

Data were generated through 106 semi-structured interviews with 105 teachers (one teacher was interviewed twice) involved in the delivery of PATHS. Interviews were conducted at four time points over two years as part of a wider study on implementation. Table 1 provides details of participants interviewed at each timepoint. This study focuses on teachers' responses to the questions about the suitability and appropriateness of the program and resources for all children, how useful PATHS was in meeting specific needs in the class, whether some groups of children were more responsive than others, whether they thought PATHS made a difference to all pupils/ some groups of pupils/ wider school population, and what pupil-level factors they thought affected implementation (see S1 for interview schedule).

## Case selection strategy

We were guided by Malterud and colleagues' concept of information power [133] in decisions around sample size, which indicates that the more information the sample holds, relevant for the actual study, the lower number of participants is needed. The adequacy of the sample was evaluated continuously during the research process, as was the feasibility of coding such a large sample [134]. We aimed for conceptual depth [135] as an alternative to saturation, assessing the range and complexity of concepts in the data, the subtleties and connections between concepts, and whether there was sufficient depth of understanding to build theory [133]. We felt it important to ensure a comprehensive and inclusive analysis that represented all of the teachers involved; this also increased the research potential of existent data in line with the principles of open research [136]. Furthermore, the matrix output provided a structure to manage a large data set; breadth can be balanced with depth [137] and a large data set can be viewed "as a whole" ([138], p.108). Previous studies utilising framework analysis have had samples of 100 or over [139–141].

## Positionality

Transcripts were primarily analysed by SH, a white woman from the UK, also a former teacher in the same region as participating schools (although in a different phase of education). SH engaged with journaling throughout the research process, reflecting on ways that her positionality, particularly professional experience, as well as gender, ethnicity, and social class, may influence interpretations, and "whether and which social systems are critically interrogated, allow or disallow observation of phenomena, reproduce oppressive structures, and determine who is visible and invisible in the data" ([142], p.vi). Whilst aiming to stay as grounded within the data as possible, bracketing off assumptions, SH's position will have impacted on themes subsequently created.

**Table 1. Teachers interviewed at each timepoint (adapted from Humphrey et al., 2018).**

| Year group | No. of teachers interviewed | Dates | Notes |
|---|---|---|---|
| 3 | 38 | November – December 2012 | 37 teachers and one teaching assistant from 22 of the 23 PATHS schools. |
| 4 | 29 | March – April 2013 | 29 of the 32 Year 4 teachers from all 23 PATHS schools. Two teachers withdrew from interview, one teacher on sick leave. |
| 6 | 20 | November – December 2013 | 20 of the 36 Year 6 teachers from 20 of the 23 PATHs schools. Only one teacher per school interviewed to reduce burden in second year of trial. Two classes no longer implementing, and one teacher left school. |
| 5 | 19 | March – April 2014 | 19 of the 35 teachers from 19 of the 23 PATHS schools (one teacher per school to reduce burden). |

## Data analysis

Data were analysed using thematic framework analysis [143]. This method is used minimally in educational research and implementation science, yet has great utility for intervention research. It is ideal for research that has "specific questions, a limited time frame, a pre-designed sample, and a priori issues that need to be dealt with" ([144], p.72). Framework analysis is not aligned with a particular epistemological position and allows for a combination of inductive and deductive approaches [145], with both a priori issues and emergent data driven themes guiding the development of the analytic framework [137]. This is important because the literature identifies factors that may influence intervention responsiveness and benefit, but this study aimed to explore the views of teachers who were directly involved in implementation. Their individual experiences can generate contextualised understandings and novel insights into the nuances of how and why an intervention may work better for some pupils compared to others. Framework analysis was also well-suited to the data because the development of a matrix provides a structure to manage a large data set, facilitating comparison across, as well as within, individual cases, and balancing depth with breadth [137]. The matrix output means that the context of each participant's views are not lost because they remain connected to their case [145] whilst also allowing a full review of the material collected [143].

Data were analysed in six stages, consistent with the framework approach: familiarisation, coding, developing a working analytical framework, applying the analytical framework, charting data into the framework matrix, and interpreting the data [145]. In the initial familiarisation stage, transcripts were read and re-read, and initial impressions of the data were noted. Data were coded in the second stage; initial codes and categories were developed using a selection of 16 transcripts (approximately 17% of the sample) from different time points and schools to ensure representativeness. Coding was both inductive and deductive: an open coding approach was adopted, whereby anything from different perspectives that might be relevant to the research questions was also coded. We then developed a theoretical framework informed by existent literature (S2). Care was taken to allow the data to dictate the categories and issues [144] and to avoid being too interpretative at this stage, so as to not obstruct the development of the later framework [137].

Coding was primarily undertaken by SH; to establish reliability 50% of the initial sample (8 transcripts; approximately 8%) was cross-checked by the rest of the team (JB, NH, and PQ). Discrepancies were resolved through discussion, coded data were re-coded where necessary, and SH coded the remaining data. The analytical framework (Table 2) was developed based on these codes, trialled with additional transcripts in an iterative process, refined, and subsequently applied to the whole dataset.

In the next stage, the coded data were charted onto a matrix, with cases (individual interviews) in rows and codes in columns. We chose to chart verbatim quotations (as opposed to summaries of comments) in cells to better retain original meanings of participants' words. The matrix is essentially a tool that charts the developing analysis [146] and is particularly helpful for synthesising large amounts of data because it presents data in a straightforward way and facilitates comparison across and within cases.

In the final stage, connections were made across cases and codes, and patterns were identified relevant to the research questions. These formed the themes and sub-themes (Table 3).

## Trustworthiness

Several steps were taken to enhance the trustworthiness of the results. Throughout the research process SH critically reflected on her positionality through reflexive journaling, as described above. The knowledge and understandings already gained from the literature, and from the main PATHS trial, were also bracketed. Care was taken to accurately effect the attitudes, beliefs, and values of the participants in the context that data were generated [143] as far as possible, with the awareness that the knowledge gained is imperfect, and qualitative research is a joint construction between researchers and participants so will always be "infused" with the researchers' subjectivity (146, p.4)

**Table 2. Analytical framework.**

| Deductive themes | Deductive codes | Inductive codes |
|---|---|---|
| Influence of gender | Specific impacts for boys<br>Specific impact for girls | Helpful for angry boys<br>Greater need among boys from socio-economically "deprived" homes<br>Helpful for "boisterous" boys<br>Girls are more engaged<br>Less need among girls<br>Gender composition of class influences boys' engagement |
| Influence of initial level of social and emotional skills | Greater benefit for those with higher level of initial skills<br>Less benefit for those with higher level of initial skills<br>Greater benefit for those with lower level of initial skills<br>Less benefit for those with lower level of initial skills<br>Level of initial skills makes no difference | More engagement and understanding among higher-attaining children<br>Boosts academic skills for children with higher level of skills<br>Not helpful for children with very low skills<br>Not helpful for children with "extreme" difficulties<br>Universal intervention not intensive enough |
| Influence of SEND and academic attainment | Differences in benefits for children with SEND<br>Differences in benefit for children with lower academic attainment | Difficulties with understanding concepts<br>Difficulties following and understanding the stories<br>Difficulties accessing resources<br>Difficulties with group work<br>Low attainment is a barrier<br>Teachers need to differentiate lessons<br>Children with SEND not expected to benefit<br>Not inclusive at all for specific types of SEND<br>Offers low attaining pupils respite from assessment<br>Higher-attaining children understand it more deeply |
| Influence of language status | Intersection with culture | Difficult for children with EAL to understand<br>Group work is difficult<br>Learning vocabulary is helpful |
| Influence of socio-economic status (SES) | Intersection with initial level of social and emotional skills<br>Intersection with gender<br>Intersection with academic attainment | Greater need among children from socio-economically "deprived" areas<br>Lower expectations of benefit<br>Influence of home life<br>Influence of parents<br>Home and school have conflicting values |

The coding process was thorough, inclusive, and comprehensive [147]; the full data set was coded to enhance confidence in the findings. To ensure that analysis provided a credible account of the data, coding was cross-checked, as described above. The matrix output improves the rigour of the research as it provides an audit trail, aiding transparency and increasing dependability [148,149]. After analysis, our results were triangulated with findings from the PATHS trial to open up a more complex, in-depth understanding [150] and enhance credibility.

## Results

### Themes

The framework analysis identified 6 main themes and 22 subthemes that reflected teachers' perceptions of [1] characteristics of children who experienced differential impacts from PATHS (RQ1), [2] their understandings of how and why those characteristics influenced intervention impact (RQ2), and [3] contextual factors influencing impact (RQ3). Inductive themes were identified to reflect perceptions surrounding the impact of parents, and the challenges of identifying intervention impact. The thematic framework is shown in Table 3.

Overall, teachers suggested that students' demographic backgrounds influenced both engagement with and responsiveness to the intervention, notably, pupils' initial level of social and emotional difficulties, their gender, and SEND status. Children with particular combinations of these characteristics experienced more pronounced differential impacts.

**Table 3. Overview of main themes and associated subthemes.**

| Theme | Associated subtheme |
|---|---|
| 1. Accumulation of advantages and disadvantages | Program needs strong socio-emotional foundations to build on<br>Inaccessible for children most in need<br>Social and emotional skills and academic skills are mutually enhancing<br>Low attainment is a barrier to engagement<br>Language barrier limits engagement for children with English as an additional language |
| 2. Differential impact among Children with SEND | Difficulties with understanding<br>PATHS provides another set of tools<br>Not helpful for children with extreme needs |
| 3. Mixed benefit for children with anger issues | Boys with low levels of anger benefitted most<br>No benefit for boys with severe anger |
| 4. Perceptions of social and emotional difficulties among children from socio-economically "deprived" backgrounds | Teachers' hesitancy to talk about poverty<br>Teachers' talk about social class<br>Cultural mismatch of attitudes and expectations<br>Helpfulness of close teacher-pupil relationships<br>Home-school conflict: parents as a barrier to social and emotional learning |
| 5. More enjoyment for girls but more benefit for boys | Perceived lack of need among girls<br>Boys are slower to respond<br>Not effective for boys with complex needs<br>The impact of classroom gender composition on SEL |
| 6. Difficulties in judging benefit for different groups | Difficulties in assessing impact: what does 'benefit' look like?<br>Reluctance to talk about demographic characteristics |

## Theme 1: Accumulation of advantages and disadvantages

**Subtheme: program needs strong socio-emotional foundations to build on.** There was a widespread perception that children needed to have a reasonably good level of existing social and emotional skills to be able to engage and benefit from PATHS. Children who did were perceived to engage with and enjoy PATHS more than others; they were viewed as more receptive to the ideas and more able to recall the strategies taught. Their understanding of emotions, self-awareness and empathy provided a foundation on which to build further understanding: "Some people have a more acute understanding of feelings and things might possibly appreciate a bit more" (Y4 teacher, School 5). However, these children were viewed as not in need of intervention because they did not appear to have any current social or emotional difficulties. Despite this, teachers described children with more secure skills using PATHS strategies to manage "falling outs" with peers (Y4 teacher, School 41) and to support other children by reminding them of the calming steps when they were upset.

**Subtheme: inaccessible for children most in need.** Children identified as most in need of social and emotional intervention were perceived not to benefit from PATHS: the children who "needed it" were the "one's that can't access it" (Y4 teacher, School 39). Teachers believed these children had issues with "access" and "engagement" because their social and emotional skills were not strong enough to begin with:

> "There's one little boy, he would find it very difficult to access because of his level of understanding. I don't think he probably gets...emotional literacy, people's point of views…so whole host of things basically...he's very difficult to stay focussed" (Y6 teacher, School 10).

According to teachers, children's difficulties with the fundamental skills of identifying and understanding emotions, self-awareness, and empathy meant that they had difficulty understanding and engaging with the PATHS stories. This was problematic because sessions were typically centred around the stories:

"We just had a child who has just gone to special and he struggled to access it but that was more to do with the needs that he had. He had severe needs in terms of emotional behaviour and even at the unit when they were doing social stories he couldn't relate to those feelings as his own. So once or twice he engaged but it's very rare. But he was an extreme" (Y4 teacher, School 7).

Children who had very weak social and emotional skills were often described by teachers as having "complex needs"; from the early stages of implementation, their expectations were that those children would not benefit from a teacher-delivered intervention. There was a perception that children with more problems needed help from an allied professional (e.g., a psychologist) rather than from a teacher-delivered intervention:

"I'm not sure how successful it is addressing his needs, but he's got a lot of problems…his problems are much more complex, that aren't going to be met by PATHS...this isn't going to answer his problems. He needs more professional help than what I am able to give" (Y3 teacher, School 10).

**Subtheme: social and emotional skills and academic skills are mutually enhancing.** Participants described children in terms of a fixed academic "ability": "You get a high average, high ability, low ability, average ability child" (Y3 teacher, School 25). Children described as "brighter" or as having a "higher ability" were also typically described as having stronger social and emotional skills, and the word "ability" was used interchangeably to refer to both social and emotional and academic skills when talking about higher-attaining pupils, as if conceptualised as a unified natural "ability". It is notable that teachers described "higher- ability" to be more engaged in lessons, answering questions "[seeming] to know what you want them to say" (Y3 teacher, School 30). Although it is conceivable that social and/ or emotional difficulties could be masked by positive academic behaviours, the children described as higher-attaining were talked about as having strong social and emotional skills, just as children with social and emotional difficulties tended to be described as lower attaining.

Children with difficulties in both the academic and social and emotional domains were viewed as problematic. Teachers conveyed a sense of frustration with what they labelled as "bad" behaviour. Children's difficulties with engagement were simultaneously attributed to "ability", "tendencies", or "personality" (Y3 teacher, School 24), as well as a choice or an unwillingness to complete tasks:

"The ones that are just got tendencies if you know what I mean. Some of my class are not only badly behaved they're really low ability as well. So some of the things I just think that they know what they've got to do but it just doesn't happen." (Y4 teacher, School 14).

Social and emotional skills and academic skills appeared to form a virtuous circle for higher-attaining children. Strong literacy skills were seen as key for understanding PATHS: higher-attaining children were thought to "grasp it quicker", whilst children with lower literacy skills were seen to "not understand all the story as well" (Y5 teacher, School 18). Stronger literacy skills facilitated pupils' understanding of the "high end" "emotional language" (Y5 teacher, School 24), as well as the concepts in the stories and "brighter" children were more able to demonstrate their deeper understanding of emotions through "eloquent" expression (Y5 teacher, School 24). In a virtuous circle, more academically able pupils were perceived to benefit more by applying learning from PATHS in literacy lessons, producing "brilliant work" (Y4 teacher, School 39). This virtuous circle may have been further enhanced by teachers' practices, through adjusting the "pitch" of their lesson to stretch "brighter" children's skills:

"We were always concerned about making sure we pitch high enough…the answers that we get in here are very eloquent, they're very well thought of, they're not just throw away comments, they're well they're well drilled in this idea of sharing ideas" (Y4 teacher, School 14).

According to teachers, higher-attaining children were typically more engaged and more confident participating in school in general, so were also naturally more readily engaged with PATHS:

> "The higher ability children generally engage more because they are the ones that are more confident speakers and more confident in their own opinions, whereas the lower ability children, fewer of them were joining in" (Y5 teacher, School 39).

Children who were more socially competent, confident, or had a better understanding of emotions, were described as more able to achieve academically because they were not impeded by social and emotional difficulties:

> "The children that I've seen using it and that it seems to be working well for are your kind of middle to higher ability children that haven't necessarily got any other sort of behavioural or social issues going on" (Y4 teacher, School 9).

Despite their perceived enjoyment PATHS, higher-attaining children were viewed as not in need of PATHS because they appeared to already have good social and emotional skills, with few or no difficulties. When talking about PATHS' benefit among these children, teachers described developed understandings of others' difficulties, and this was believed to help them "cope" with, rather than being "disturbed by" the behaviour of children who do have difficulties (Y4 teacher, School 29), as if to contain the disruption so that the other pupils could continue with classwork:

> "Almost like a support group that's still using PATHS to help the other children to understand [his] behaviour, which is quite extreme at times, and helping to come up with coping strategies for when that's happening and then understanding that [his] thinking at times like that is alternative to theirs" (Y4 teacher, School 9).

**Subtheme: low academic attainment is a barrier to engagement.** Similarly, children who were described as having weaker social and emotional skills were also typically described as less academically "able". However, social and emotional skills and academic skills appeared more distinct among this group than in higher-attaining children, perhaps because social and emotional difficulties manifested in ways more visible to teachers, such as anger, or behavioural or social difficulties, and therefore were talked about in more depth. For lower-attaining children, social and emotional skills and academic skills formed a vicious circle, through which disadvantage in one domain perpetuated disadvantage in the other.

Low academic skills were perceived as a barrier to engagement with PATHS. Lower-attaining children were described as having to "play catch up most of the time" (Y5 teacher, School 24). To benefit, children needed to have a certain level of academic skills because a good level of literacy was needed to understand the language, as well as the concepts:

> "It doesn't matter how information you give them about a feeling, they struggle with the concept anyway...some of the language - if I was to follow that script completely would be way above them. So, you'd have to adapt the language" (Y3 teacher, School 18).

Again, teachers talked about the PATHS stories being difficult to access for lower-attaining pupils because of lower literacy levels. Teachers described children unable to maintain attention during the stories and said that the writing tasks were too difficult:

> "Some of the stories are a little bit long and without me beforehand spending a bit of time reading it all through and doing any changes…some of them haven't got the attention span and they will just drift off, kind of thing" (Y5 teacher, School 43).

Since the stories typically introduced the content of each session, having attentional difficulties during the story reading impacted negatively on engagement with the rest of the session because pupils "have to have enough content to get to the issue" (Y5 teacher, School 43). Thus, pupils who already found it more difficult to maintain attention were then further disadvantaged in accessing the rest of the session. It could also be the case that these children struggled with attention and "switched off" because they found it difficult to understand/engage with the emotions; there was an awareness that there may be multiple factors involved:

> "He finds it really difficult to concentrate in lessons…sometimes he is not motivated because he is not interested, because he doesn't understand and because he doesn't know the concepts are maybe going over his head" (Y6 teacher, School 30).

Teachers talked about the challenge of differentiating materials to make them more understandable and engaging for lower-attaining children "without losing the purpose of the lesson" (Y5 teacher, School 10). Teachers also described using only some of the materials or delivering only parts of sessions with these children, for instance, focusing only on the feeling cards to learn emotion words. Whilst this was perceived as beneficial for these children, it meant that they were not given the same opportunities to benefit as their higher-attaining peers:

> "Some children might ask what words mean sometimes...if it is a word they're not sure of it would just be improving their vocabulary anyway so it's words that they should be accessing so it's helpful in that sense" (Y6 teacher, School 9).

Teachers also described the negative impacts of ability grouping within classrooms for lower-attaining children:

> "They know their ability groups children, it is awful. As a teacher sometimes you think they're not aware of it but they are…No matter how you try and mask it with shapes or colours of groups…they know" (Y3 teacher, School 18).

This was said to make children feel "judged" and "boxed in because of their need" (Y3 teacher, School 18). However, despite difficulties with the content of the program, one benefit for lower-attaining children was that PATHS lessons offered respite from the pressure of academic assessments and progress tracking:

> "They seem to know when it's PATHS and they know there's not that same kind of "right this is what you need to achieve" so therefore they just become a bit more just self-motivated...they know there's not that pressure...not being marked against anything" (Y3 teacher, School 25).

Teachers said that this respite from progress tracking meant that children could be sat in 'mixed-ability' groups for PATHS lessons, but this often meant that those with difficulties were left out:

> "They fall down when they have to work in groups. It does worry me that are the more able children are they doing most of the work?...It's hard to assess" (Y5 teacher, School 17).

**Subtheme: language barrier limits engagement for children with English as an additional language (EAL).** An important contextual factor was whether children's difficulties with literacy were owing to English being an additional language for them. According to teachers, children with EAL found the sessions "difficult"; they struggled to understand the stories and had more difficulties participating in group work with others because of the language barriers. EAL children were described as less "focused and engaged in it purely because of the language barrier" (Y5 teacher, school 24). Despite being viewed as "quite vital for development" (Y5 teacher, School 13), PATHS was not considered a priority for EAL children. Because of this, and because teachers did not expect such children to be able to engage, PATHS was

used as a literacy intervention to increase EAL children's vocabulary. As with children with lower literacy skills, learning emotional vocabulary was perceived as beneficial for EAL children, but there was a sense that this was the extent of the benefit and these children were not able to access the social and emotional program in its entirety:

> "The children with EAL, knowing a word other than sad or happy…they've got the word banks there and we've read through them...I've picked a word, they've looked it up and given me the definition" (Y5 teacher, School 41).

**Theme 2: differential impact among Children with SEND**

**Subtheme: difficulties with understanding.**  Teachers perceived PATHS to have less or no benefit for pupils with SEND because the content was too difficult to understand. How much these pupils were able to access appeared to depend on adaptations made by teachers, additional adult support, and the type of SEND.

According to teachers, Children with SEND had difficulties reading the vocabulary and understanding the language in the stories and materials. Listening to the teacher-read script was described as "way above them" (Y3 teacher, School 18). All teachers described having to make adaptations to the language for Children with SEND to access PATHS. Although the 'feeling faces' were considered helpful because they did not require children to read words, teachers said that "it doesn't matter how information you give them about, say a feeling, they struggle with the concept anyway" (Y3 teacher, School 18). Additionally, the "problem-solving steps" were believed to be too complicated for Children with SEND and "harder to recall" (Y5 teacher, School 17):

> "The three steps to calming down confuses them, the posters are a little bit confusing, because you have got the control signals and then another poster that unwraps if you like the red one, the red signal and I think that does get them a bit confused" (Y4 teacher, School 32).

Pupils with SEND were said to need extra adult support to provide "extra explanations", help pupils to "stay focused", and "promote, provoke them into action" (Y4 teacher, School 25). Teachers described how pupils with SEND already had difficulties with reading, learning vocabulary, understanding, and concentrating, and that adapting, or differentiating, all activities across the school curriculum was standard practice. Pupils with SEND were described as being "on the outside" and "struggling to engage" (Y3 teacher, School 39) with the rest of the class in general. Even before implementation, teachers' expectations were that these children would not be able to fully engage with PATHS, and certainly not without adaptations. Pupils' difficulties with PATHS were attributed to their SEND, rather than a failure of the intervention: teachers believed that it was their "issues" that "would stop them accessing PATHS" (Y4 teacher, School 39). There was a belief that PATHS was not working, and would not work, because of "the nature of the children" (Y5 teacher, School 39). There was also a sense that, because these pupils were viewed as being outside of the mainstream curriculum, teachers did not see PATHS as relevant to them, since they already had an identified 'need' and associated support plan.

**Subtheme: PATHS provides another set of tools.**  A contextual factor that influenced the impact of PATHS among Children with SEND was the nature of other forms of support they received. Teachers considered PATHS as "useful" when it was viewed as better than the existing support or intervention, or as a substitute where the desired support was not available (e.g., children with undiagnosed difficulties who were not eligible for support or medication). In these situations, PATHS provided "another set of tools to work with" (Y4 teacher, School 36). However, where PATHS was seen as competing with existing support that was well-established in the school, it was perceived as clashing and confusing for children, negatively impacting on engagement:

> "He's not accessing it because for a couple of reasons, some of the other adults that work with him for interventions don't use it so if he goes to this person they might be doing something slightly different to this person, and this person, so it's about everybody's not got that consistency" (Y3 teacher, School 39).

How additional support was organised within schools also impacted pupils' engagement with PATHS: it was more difficult for pupils to engage if they missed part of the lesson due to being timetabled for other support at the same time:

"They come and go, some of the special needs children, then back in and didn't know what we were doing...she is not very well behaved anyway, so she just didn't listen, but she is like that anyway" (Y3 teacher, School 32).

**Subtheme: not helpful for children with extreme needs.** The nature of pupils' SEND was an important contextual factor that influenced the impact of the intervention. Teachers perceived PATHS as not effective for children with autistic spectrum disorder (ASD) because it was "too abstract" (Y5 teacher, School 6) and because it relied on children having a certain level of social skills. Some of the strategies assumed a neurotypical perspective, for instance, "the one where someone calls you a name, you call someone something back to make it funny...a boy with Asperger's definitely wouldn't be able to grasp that" (Y5 teacher, School 18). According to teachers, children with ASD also struggled to engage because of difficulties with identifying and showing emotions; tasks involving matching "faces and feelings together" were very difficult (Y3 teacher, School 10).

Similarly, children with more "extreme" SENDs, such as pathological demand avoidance (PDA) and oppositional defiance disorder (ODD), were perceived not to engage because their difficulties with emotions were viewed as too severe. Although children described as "extreme" were discussed as if they were a "rare case", they were a common feature of teachers' conversations. When such children did show engagement, this was dismissed because it was viewed as "very rare" or "hit and miss" (Y4 teacher, School 9). Teachers dismissed any notion of benefit for these children because they were seen as unchangeable and impossible to reach:

"He will not engage at all with this program. He won't engage with very much…Apparently he's oppositional defiance disorder which I think is a title for being a naughty boy, won't do as he's told, or rather pleases himself which staff he'll behave for…So I think it's just his nature" (Y5 teacher, School 36).

Again, there was some labelling and attribution of children's behaviours to "nature" or unwillingness to engage.

### Theme 3: mixed benefit for children with anger issues

**Subtheme: boys with low levels of anger benefitted most.** Teachers perceived a subgroup of children with anger issues to gain particular benefit from PATHS. These children tended to be boys, and there was a perception that this group needed social and emotional support because they lacked the skills to manage their feelings. There was also a perception that boys generally tended to "suppress their feelings and not necessarily look into them and understand", and that "normally it is more boys that fall into the behaviour groups" (Y6 teacher, School 9).

PATHS was perceived to be especially beneficial for boys who had a moderate level of anger, described as "a temper". According to teachers, the learning of steps in the PATHS curriculum enabled those children to "think about their anger and the consequences", which slowed down the expression of anger and helped them to calm down (Y4 teacher, School 36). The "motor responses" such as linking fingers were seen to be effective with these children because they enabled the child to externalise feelings in a positive way, for instance, by physically linking fingers rather than "stomping around" (Y3 teacher, School 5). The focus on strategies for calming down also helped engage angry children because it was less stigmatising; for children who were used to being problematised and punished, an approach whereby anger was normalised and accepted was believed to be more helpful:

"Because they get things explained to them about how they're feeling without it being directed at them so it's quite a safe environment for them to start thinking well perhaps I can talk about it" (Y3 teacher, School 5)

**Subtheme: no benefit for boys with severe anger.** PATHS was perceived as not effective for children with more severe anger. Those children tended to be boys from what teachers described as "deprived" households; they were viewed to be unable to use the strategies at the time of experiencing anger because their emotions overpowered their ability to recall them, and teachers described trying to teach these children as a "battle". PATHS was described as a "fast fix" (Y6 teacher, School 7), not intensive enough to treat the root of the issue or prevent the anger in the first place:

> "Not fully of benefit because he still gets angry, it doesn't stop him from getting angry, just helps to calm down. The damage is already done because he's already lost his temper, he's already got upset" (Y3 teacher, School 25).

Among these children, anger was thought to be deep-rooted and stemmed from home or early childhood, or related to 'innate' difficulties with other social and emotional skills, such as self-awareness and empathy. Like other children with lower social and emotional skills, teachers felt that children with severe anger were unable to benefit from PATHS because they were too lacking in the foundational skills of empathy and self-awareness:

> "I think you just want a fast fix and I think it's quite tough for me to say it's not having the impact…I don't think there is any. I really am fighting nature a lot of the time...if they were listening long enough to actually get the story, they can't relate themselves to have the empathy for the characters within it. And then make that contrast with themselves. And then to apply it to think "oh I can change". They don't make that connection" (Y6 teacher, School 7).

### Theme 4: perceptions of social and emotional difficulties among children from socio-economically "deprived" backgrounds

Teachers believed socio-economic "deprivation" was at the root of most children's emotional and behavioural difficulties. It was viewed as the most salient demographic influence, "rather than the ethnicity or something like that" (Y4 teacher, School 24). Teachers said children from "deprived" homes had "lots of difficulties" (Y3 teacher, School 17), and this was "obviously because of where they live and things" (Y3 teacher, School 5). These difficulties were described in terms of behavioural and conduct problems, anger, and more negative emotions such as anger, stress, and worry. This group of children were described as having more to cope with, through having "tough lives" and becoming "independent at a much younger age" (Y6 teacher, School 9). They were described as having generally "more feelings" (Y3 teacher, School 43) than other children, and, therefore, needed additional support to cope.

**Subtheme: teachers' hesitancy to talk about poverty.** There appeared to be unease and a tentativeness when talking about poverty and socio-economic "deprivation", with teachers using hesitations and vague terms such as "area" to refer to the socio-economic context of the communities served by the school:

> "There's a lot sort of around this area that is quite you know kids have a lot of difficulties … a lot of things to do with erm family life can be really tough for some of them and obviously that comes out in school" (Y4 teacher, School 27).

The use of ambiguous phrases such as "certain children" and "issues at home" were possibly to avoid revealing sensitive or confidential information, or to avoid seeming judgmental or stigmatising. Talking about the effects of poverty seemed taboo and was commonly reframed as being associated with "family life" and, therefore, children's private lives. Whilst teachers talked freely about individual children whose difficulties were relating to SEND, there was a hesitancy to talk about individuals in relation to socio-economic deprivation. Teachers used phrases such as "let me just think about how to word it" (Y3 teacher, School 13), appearing aware of their own positioning:

"Some of the things like the calming down isn't not all of them need it but you wouldn't want to single one of them out and say "you need to use this"" (Y6 teacher, School 2).

There appeared a presumption that children from "deprived" areas all had difficult home lives, and an awareness that children would feel stigmatised as a result.

**Subtheme: teachers' talk about social class.** Although it was clear that there was a large proportion of children experiencing difficulties because of poverty, teachers' preconceptions of social class appeared to influence their perceptions of children's social and emotional skills, and their expectations of how pupils would benefit from PATHS. Teachers described how class differences in cultural norms between themselves and pupils made them uncomfortable when children talked about their families:

"There is a whole group of children here who quite like to shout all their family problems to the rooftops and I think because their families do that's what they've learned to do. And that's something I've felt a little bit perhaps a bit middle class and uncomfortable around" (Y4 teacher, School 43).

Although PATHS encouraged children to talk about their worries, there appeared a devaluing of this practice for these children. Not talking openly about problems, and positive characteristics such as empathy and calmness, were associated with middle class children: it was seen as surprising that a school that was not a "middle-class school" would have "a nice empathy" and "calm" children (Y3 teacher, School 17). In contrast, poor social skills and behavioural problems were associated with children from lower class backgrounds:

"We've got a lot of…I don't know, sort of class distinctions here…a lot of our families, they're not even working class, they're sort of you know, parents who don't work…and a lot of the children as they progress up to high school...have got into trouble or they don't last very long at high school" (Y3 teacher, School 41).

Children from lower social classes were discussed in terms of skills deficits, whereas children from more affluent middle-class homes were perceived to have better skills by default. They were believed to be less in need of social and emotional intervention because of an absence of observable difficulties and a protective middle-class home. Teachers associated anger and poor emotion regulation with lower class children, and because of the emphasis on calming down and managing anger, PATHS was viewed as not relevant for middle class children:

"The children that we have here are mostly from…you know, nice area, well established…don't sort of want for much if you know what I mean, so…I'm not sure they're going to fully benefit from the whole range of skills" (Y4 teacher, School 6).

There was, however, an awareness that middle class children could go on to develop difficulties later in childhood if they have to deal with challenging circumstances or socialise with children from "different backgrounds" (Y4 teacher, School 24). Yet, whilst teachers framed talk about deprived children getting into trouble in high school as a consequence of their social and emotional skill deficits, expectations relating to middle class children's future difficulties in high school were not framed in terms of innateness or skills deficits, but thought of as a result of the change in environment, or because they had not been "equipped with the tools" (Y6 teacher, School 17):

"We live in an area which is…fairly affluent…if anything comes into the lesson or into their life that changes, that puts a spanner in the works they're not very good at coping with those situations… not everybody is living lives like some of the children are in our school…so it's bringing that greater diversity" (Y4 teacher, School 24).

For children from "deprived" backgrounds, however, PATHS was described as helpful through providing opportunities to talk about problems from home; there was a belief that these children were not afforded such opportunities at home because their parents typically had "lots of children at home" and were stressed, arguing or generally "not that interactive with the children" (Y3 teacher, School 38):

> "There's a lot sort of around this area that is quite…kids have a lot of difficulties…a lot of things to do with, erm, family life can be really tough for some of them...PATHS gives them that time to talk about it" (Y4 teacher, School 27).

However, beyond this, there was the sense that PATHS had little impact on such children. Because their difficulties were perceived to be caused by their home life, they could not be prevented by anything taught at school. Children would go home after school to face the same problems, and their home life was believed to exert a more powerful influence than school. Interactions with parents outside of school were perceived to have a huge impact: "if they have a bad morning with their parents, they're coming into school swearing about their parents. Then what we talk about in PATHS isn't necessarily working" (Y4 teacher, School 39).

**Subtheme: cultural mismatch of attitudes and expectations.** Teachers said that much of the PATHS content lacked cultural relevancy for children from "deprived" backgrounds and inner-city areas because it was too "middle class" (Y6 teacher, School 39) and "suburban" (Y5 teacher, School 39):

> "Quite a lot of the stories aren't relevant to our particular children because of their background…stories about going to shops and choosing new clothes and that doesn't happen very often for the kids in our school. Going on holidays and that doesn't really happen…quite a lot of the issues in the stories is to do with money and they don't have disposable incomes in the same way the characters in the story do. The other is to do with travelling. Quite a lot of them have never left this small area of Manchester…They don't have gardens, the stories talk about going out and playing and they're not allowed to, they stay in in the evenings" (Y4 teacher, School 39).

Teachers described having to make adaptations to "about 80 per cent" of the stories to "make them a bit more relevant for their lives" (Y4 teacher, School 39) so that these children could engage with the meanings. Some of the stories were viewed as inappropriate or harmful for children experiencing certain adversities (for instance, bereavement or removal from the family home). There was also the view that more broadly, PATHS was not helpful to children from "difficult backgrounds" because it did not consider that some children may not have the same type of stability or family depicted in the stories:

> "There are children from particularly difficult backgrounds or situations that I think some of the questions are not appropriate, because they are just in a difficult place at the moment…certain things, I don't think that is helpful to that child just now...I think some of the wordings and things funnily enough assume the children seem to be in quite a solid place" (Y4 teacher, School 32).

Even after making adaptations, teachers commented that the American attitude of having a "positive spin on everything" was unrealistic and "cheesy" for many English/British children who had difficult lives, and therefore may have been less effective for children from "deprived" urban backgrounds.

**Subtheme: helpfulness of close teacher-pupil relationships.** Teachers described children with social and emotional difficulties as engaging more if they were able to establish a shared social and cultural identity with pupils:

> "I threw myself into this the other week, giving myself as an example and he was "Miss I didn't know that" you have got to get to the heart of it...I throw a bit of me in, or maybe about my son, I come from one of the biggest families in [the area]...I can throw a bit in and they love it...Because it makes sense with them" (Y6 teacher, School 43).

This helped to make the PATHS stories more relatable for children, as well as contributing to closer teacher-pupil relationships, which were seen as important for the effectiveness of PATHS. Close and trusting relationships within the class, and feeling "comfortable with the teacher" (Y5 teacher, School 41) were thought to help children feel "safe" to share problems or personal stories. Teachers believed this was necessary for the discussion component of PATHS to be meaningful:

> "I had to build up the relationships first before I could get that personal discussion going with them...how they are seeing me and how they see each other and how we all interact together. I think I have built generally quite good relationships with them...I think that has helped" (Y5 teacher, School 39).

Teachers also believed that having a good relationship with pupils meant that they could gain more knowledge about children's lives when they shared problems. This enabled teachers to personalise PATHS lessons for certain pupils facing difficulties, to increase benefit, for instance, by "saying certain words within the lesson" that will "help a child" (Y6 teacher, School 43).

PATHS also provided opportunities for staff to talk to pupils on a more personal level "outside the madness of the curriculum" (Y6 teacher, School 7), so in a virtuous cycle, could strengthen relationships between teacher and pupils. Developing close and trusting relationships was also important for helping children with social and emotional difficulties talk about problems during PATHS because it could open referrals for more intensive support:

> "It flags other stuff up for me then and I might be able to refer him on or get him to self-refer to 'Place to Be' or, more often than not he doesn't want to, he just wants to let me know" (Y4 teacher, School 43).

**Subtheme: home-school conflict: parents as a barrier.** When talking about increased need among "deprived" children, parents were the focal point and parenting was described as a "big issue" (Y3 teacher, School 13). Parents were perceived as the link between deprivation and their children's difficulties and skill deficits. Teachers offered insights into "deprived" pupils' home lives, in which pupils' difficulties were described in terms of 'deficits', attributed to the absence of positive role models, and a failure of parents to teach children "manners", or "how to express themselves emotionally", or "how to be social" (Y3 teacher, School 19). Teachers saw themselves as having to counter negative influences from parents:

> "I like teaching children strategies to support when they're feeling frustrated…because it's something that maybe hasn't been modelled before…if you are from a chaotic background, or where the only way to react is by shouting or by…or if your dad is telling you to push, punch, shove to get what, which does happen" (Y3 teacher, School 10).

Teachers' tentative language indicated some level of speculation about pupils' lives, and pupils' difficulties were attributed to their home lives despite teachers being "not sure" about what children "experienced at home" or "what [they've] seen" (Y3 teacher, School 1). Assumptions were made that "it's families basically" and that these children have "probably gone through loads before they even get to us" (Y6 teacher, School 25). Even less severe difficulties were attributed to parents, for instance, "tiredness" was assumed to be caused by a problematic home life (Y3 teacher, School 43) and worry about spelling tests was believed to be caused by "pressure from home" rather than from school (Y3 teacher, School 17).

There was an undertone of blame and criticism in teachers' talk about parents. For instance, although there was an awareness that "deprived" children were not afforded the same opportunities or resources as their wealthier peers, this was framed as a criticism of parents rather than a societal problem:

> "It's an area of sort of socio and economic sort of deprivation. Some children do not have the opportunities at home that many areas might provide. Sort of…they have become quite independent at a much younger age so they're not necessarily given opportunities from their parents that they would get in other areas" (Y6 teacher, School 9).

These children's issues were viewed as too deep-rooted for PATHS to reach because particular behaviours were already "embedded" in early childhood. Teachers expressed beliefs about how emotions were socialised by parents, and parents were seen as a significant barrier to social and emotional learning, with the perception that the values, attitudes, and behaviours demonstrated at home conflicted with those of school, at the detriment to children; they would "undo" anything learned in PATHS sessions:

> "I think you've got a lot of the learnt behaviours already in place so they can tell you what they should have done in retrospect, but it's quite hard at this stage...a lot of stuff's embedded in them already, act out, strike out, hit, natural reflex, call you a name back… it's parents as well...we get the parents and they say I've told them to hit them back and it's like you're wrestling with that as well, they're getting told one thing by one, one by another and frustrating" (Y4 teacher, School 25).

Teachers also believed that parents may object specifically to the PATHS curriculum for personal, cultural, or religious reasons, so omitted some sessions from the program:

> "One involved being offered alcohol and we just didn't think it was appropriate for our intake of children. So I think it would open a bit of a can of worms…it could have got misconstrued if children had gone back saying we did a story about children drinking alcohol. It isn't that, but parents wouldn't understand" (Y6 teacher, School 17).

These beliefs appear to have been formed from more general expectations of parents relating to the "home-school barrier" (Y5, School 13), since teachers reported not using the home resources or sharing information about PATHS with parents. Doing so was thought to be "potentially a waste of time and resources" because teachers believed that the parents of children with social and emotional difficulties did not value education, or likely experienced the same difficulties as their children and would "not understand" the homework or the intervention. There was also a belief that attempts at involving parents had the potential to cause conflict:

> "Some people might think it is a bit intrusive...It would be quite rude, they might think that is a bit too intense, a bit personal if there is...things going on in their home lives" (Y6 teacher, School 43).

Thus, the children who most needed support at home were those least likely to receive it. Even if parents did understand the homework, teachers were worried they would "moan and say there's too much being asked of them" (Y4 teacher, School 25), and if they did complete the homework that they may not properly "engage" with it. The "home learning barrier" was as described as a "hard one to get over" (Y5 teacher, School 13).

### Theme 5: more enjoyment for girls but more benefit for boys

**Subtheme: perceived lack of need among girls.** Teachers described girls and boys to respond differently to PATHS. Girls were perceived to be more engaged, more responsive, and express more enjoyment in PATHS lessons than boys. However, boys were perceived to gain more benefit from the intervention than girls. The differences in responsiveness and engagement appear to relate to the perceived differences in boys' and girls' social and emotional skills and difficulties, their willingness to talk openly about feelings, and reluctance to be seen by peers talking about feelings.

According to teachers, girls were more responsive because they tended to have stronger social and emotional skills and fewer difficulties. They were described as "generally much more mature" (Y4 teacher, School 18), "more mellow than boys" (Y3 teacher, School 17) and subsequently more able to "connect with the feelings and emotions" (Y4 teacher, School 43). Girls were also seen as more comfortable sharing their feelings and stories with others. Despite this, there was a perception that girls did not benefit from this because they already had a good understanding of emotions. There

was also an awareness that girls were generally more motivated across the school curriculum; they were "pleasers", so their participation in PATHS may be driven by a motivation to "please" the teacher and provide an acceptable answer, rather than reflecting deep engagement with PATHS:

> "Your girly girls really want to please and really want to give you examples. In fact you have to try and shut them up sometimes. They want to give lots and lots and lots of examples. Sometimes some of the boys are a little bit more reluctant but then some of them will surprise you" (Y4 teacher, School 7).

Teachers perceived the friendship element of PATHS to have noticeable impact among girls, and reported fewer "name calling" and "fallings out" among girls. As was the case for children with stronger social and emotional skills, teachers noticed girls using the strategies in social situations.

**Subtheme: boys are slower to respond.** Different patterns were reported for boys. They were perceived to be less engaged, less responsive, and slower to respond, than girls; "a bit harder to get through to" (Y3 teacher, School 24). Despite this, teachers believed boys benefitted more from PATHS than girls. There was an awareness that *appearing* to lack engagement did not necessarily mean a pupil was not engaged, and teachers described boys feigning disengagement to mask their unease with talking about feelings:

> "The girls enjoy it more than the boys, I don't know whether that is because the boys won't admit it…often as soon as we talk about feelings, the boys will make a joke and say it is daft or whatever and I think that is a lot of bravado – I think that maybe underneath they enjoy it, because they certainly contribute, once they get the giddiness out of the way, they will contribute. So I think they may enjoy it, but they find it more uncomfortable to admit that, where the girls will say that it is good and that it is helpful" (Y6 teacher, School 39).

Another view was that boys may have appeared to lack engagement because the PATHS strategies were seen as "baby-ish" or otherwise socially unacceptable among their peers:

> "The children are quite streetwise and maybe they don't feel comfortable with doing that approach...some of them… would be conscious about their image of doing that" (Y5 teacher, School 13).

PATHS was perceived to be particularly beneficial for boys with relatively low-level difficulties, such as those described as "immature", "bossy" or "alpha males" (Y4 teacher, School 7), or those who "get cross". Teachers talked about "boisterous behaviour", "rough play", and minor conflicts, typically at playtime, over football or Lego figures. These boys were perceived to benefit from the strategies to help manage anger or "control their temper", and teachers noticed fewer conflicts between boys since implementing PATHS.

**Subtheme: not effective for boys with complex needs.** PATHS was perceived as not effective for boys with more complex needs, who were often at the intersection of multiple forms of disadvantage:

> "There is a need particularly because of our demographic…children aren't as educated in understanding the way that they feel, particularly boys. Maybe the common thing is for them to suppress their feelings and not necessarily look into them and understand. So there is this sort of missing link" (Y6 teacher, School 9).

For this subgroup of boys, impact was perceived to vary depending on the influence of other intersecting characteristics, such as the nature of their difficulties and/or SEND, their academic, social, and emotional skills, and their home life. According to teachers, the children with the lowest social and emotional and academic abilities, the most difficulties, and the more "extreme" SEND, all tended to be boys.

**Subtheme: the impact of classroom gender composition on SEL.** The classroom gender composition was believed to influence the responsiveness of boys through the "class dynamic" (Y3 teacher, School 17). Teachers believed that there was more problematic "boisterous" behaviour in classes that had many more boys than girls, and "behaviour issues" were perceived as inevitable if there were a large number of boys in the class, "especially at playtime with football and things" (Y4 teacher, School 30). Boys in these classes were perceived to "take ages" to respond to PATHS (Y4 teacher, School 6), but having more girls than boys in the class was seen to make a "big difference" (Y3 teacher, School 17). Being in a predominantly female class was perceived to be beneficial for boys' behaviour, since girls were generally viewed as having secure social and emotional skills by being undisruptive and "calm". Boys more readily adopted the PATHS strategies for calming down, or were perceived not to need to use them, when in classes with fewer boys and more girls:

> "There's a lot of girls which makes a big difference. There's only nine boys so it's a different dynamic…For other years I would probably think it's very crucial to teach the calming down, whereas this class I haven't had to use it" (Y3 teacher, School 17).

**Theme 6: Difficulties in judging benefit for different groups**

**Subtheme: difficulties in assessing impact: what does 'benefit' look like?.** Whilst teachers perceived differential impacts among certain groups, those perceptions were based on behaviours that were easily observed, such as children raising hands to answer questions, verbal and written responses to questions, participation in group work and role plays, visible use of the PATHS strategies, and changes in externalised behaviour. There was an awareness that it was "hard to assess" whether some children are engaged or "switching off" (Y5 teacher, School 17), and even not possible to judge the impacts for children "aren't as verbal about what they're doing" (Y3 teacher, School 26), or who respond in less visible ways. Indeed, one thing that both teachers and pupils appeared to like about PATHS was that pupils' learning was not assessed as it was in other curriculum areas:

> "We've got so much to do and it's that you've got to do this, you've got to fit this in, you've got to self-assess and der der der, it's just nice to just sit there and talk to them about how they're feeling" (Y3 teacher, School 5).

**Subtheme: reluctance to talk about demographic characteristics.** Overall, there was a reluctance to talk about demographic differences, with a preference for focusing on children as "individuals":

> "I don't normally just focus something on one particular group…I call out names and I try not to…I don't focus on the group I just try and focus on something I see that's inappropriate" (Y4 teacher, School 14).

What appeared to be considered as good intentions to treat children as individual unfortunately meant that patterns among children with certain demographic characteristics, such as those relating to ethnicity, as described above, were not considered. Instead, children were generally talked about in terms of academic attainment and grouped as "brighter ones", "less able children", or "lower ones".

## Discussion

We set out to understand how teachers understood, described, and explained the variations in impact of the PATHS curriculum on students, and the relationships between contextual factors and students' learning in the PATHS curriculum. Our analyses indicate that teachers believe some subgroups of children respond and benefit more than others: boys with a moderate level of social and/or emotional difficulties were believed to benefit the most, whilst children with SEND and boys with

more severe social and/or emotional difficulties were thought to benefit the least. These subgroups comprised children with multiple intersecting characteristics, the interaction of which influenced the extent of their benefit from the intervention.

Participant responsiveness is a key element of implementation and has been defined as "participants' level of enthusiasm for and participation in an intervention" (84, p.23). It is often operationalised as number of sessions attended, as well as retention, homework completion, engagement, and satisfaction [84,151,152]. Our findings demonstrate that a child may be physically present during a session of intervention delivery, but, according to teachers, may not engage in a meaningful way. Intervention level factors may impact negatively on child engagement, such as inaccessible content or materials, or cultural mismatch. Our analyses also suggest that teachers' beliefs about pupils' responsiveness and/or engagement may be influenced by biases in respect to the social class, gender, and SEND status of pupils. These biases may lead to differential expectations and implementation behaviours towards different groups of pupils, and may in turn affect pupil engagement and responsiveness.

Our analyses also indicate that teachers may be more able to assess intervention impact among pupils with externalising behaviours over pupils with internalising symptoms. Contextual factors were perceived to influence intervention responsiveness and impact: boys were perceived to be more responsive in classes with a higher ratio of girls to boys; and PATHS was perceived to have a greater impact when teachers and pupils had closer relationships.

### Perceptions of impact among boys with a degree of social and emotional difficulties

The subgroup believed to benefit most were boys who had a degree of social and emotional difficulties, provided that they had a secure enough level of literacy, academic skills, and understanding to access the program. Teachers descriptions of boys' behaviour aligns with a wide body of research reporting that externalising behaviours are more common among boys than girls in primary school. However, since externalising behaviours are more visible and more disruptive in the classroom [153], it may be that they were more noticeable to teachers. Thus, teachers may have been more able to identify pupils with difficulties, and consequently observe improvements, compared to those pupils who internalised their distress. Previous studies of teachers' ability to identify mental health concerns among primary school aged children have found that they have more difficulty accurately identifying internalising problems than externalising problems, and that they perceived externalizing problems to be more serious and more concerning [154–156]. Children with externalising problems are more likely to be referred for intervention than those with internalising problems [155]. Furthermore, the main trial in which the data were generated reported no significant differences between children in PATHS and usual provision schools in respect of teachers' reports of their internalising symptoms, but did find a statistically significant improvement in *children's* self-reports of psychological well-being [125]. Thus, it could be speculated that teachers were less perceptive of the impact of PATHS on children's internalising symptoms. However, the main trial found a significant increase in prosocial behaviour and a small reduction in emotional symptoms among children identified as high risk (those with elevated symptoms at baseline) in both child and teacher reports [125]. This aligns with our finding that teachers reported benefit for children with some degree of difficulties, but suggests that teachers *were,* to some extent, able to identify internalising difficulties and improvements. It would be useful to ascertain teachers' understanding of emotional symptoms, since it could be the case that teachers understood these mostly as feelings of anger, which were well-talked about.

### Perceptions of differential impact for boys and girls

Our analyses also suggest that, when talking about impact for boys and girls, teachers' understandings and expectations of children's behaviour may be based on socially constructed views of gender. Boys were viewed as prone to behaviour problems and described as "Alpha males"; in contrast, girls were "pleasers" (see *Theme 5* in results section). These could be reflective of the differences in the socialisation of emotion for boys and girls evident in the education system [157], as well as gendered expectations relating to behaviour [158] and achievement [98,159]. In other studies, teachers reportedly

perceive girls as more motivated and boys as displaying more troublesome behaviours [160] and higher levels of external-ising behaviour than girls [161]. Teachers' own assumptions around the gendered nature of emotionality have also been found to influence their practices specifically when delivering SEL interventions [162]. For instance, Evans' qualitative study of the delivery of a SEL intervention in a secondary school found that teachers interpreted the same behaviours differently for male and female students, and that this influenced their behaviours, or "micro-practices" towards students according to their gender: teachers viewed boys' passivity as a sign they were "taking it in", whereas when exhibiting the same passivity, girls were pressured to make emotional disclosures to "overtly demonstrate their social and emotional competency" (162, p.198). Since our study data do not include observations, we cannot conclude whether these apparent and beliefs and expectations impacted on teachers' practices when implementing PATHS. However, this warrants atten-tion, because if teachers' practices are indeed influenced by gendered expectations of behaviour and emotion expression, this could be counterintuitive to SEL. If teachers hold expectations that boys are prone to anger or disruptive behaviour, this could result in self-fulfilling prophecies, which can limit children's opportunities [163] (this will be discussed below). Likewise, if teachers hold expectations of girls as being less in need of SEL, their difficulties may be overlooked and resources could be diverted away from them.

In our study, girls *were* generally viewed as not in need of social and emotional intervention because they appeared engaged in the classroom and lacked observable social and/or emotional difficulties. Although some national survey data generated closer in time to the PATHS trial does indicate that at primary school age, social and emotional difficulties were more prevalent among boys [70], lower prevalence among girls is not the same as no prevalence. Furthermore, this data also relies on teacher reports, which could be subject to similar biases to those noted above. It is interesting that this trend is reversed when data are drawn from children's self-reports [70]. Whilst emotional difficulties have been found to peak in mid-adolescence [164], it may be that these are less noticeable to teachers in younger girls. Indeed, high levels of emotional symptoms have been found among younger girls in child self-report data [165], and more recent data based on parent and child-reports indicate that the prevalence of social and emotional difficulties is similar for boys and girls of primary school age [166].

Even if it was that girls did not have any social and emotional difficulties, the beliefs that PATHS was not relevant to them because of this suggests that there may be misconceptions in relation to the aims of USB SEL interventions: teach-ers may expect that such interventions are a treatment for existing problems rather than to *prevent* future problems that have not yet emerged [1]. If teachers believe that USB SEL interventions are not appropriate or relevant, this may nega-tively impact implementation behaviours [167]. It is important that SEL training includes the theory and principles behind the preventative approach, since a deep understanding of program goals is necessary for implementation quality [86].

## Intersection of gender, poverty, and social and emotional difficulties

Our analyses suggests that teachers perceived socio-economic deprivation to have a particular impact on the social and emotional development of boys. Boys from socio-economically deprived homes were believed to experience difficul-ties in coping with family and home-related stress because they have been socialised by parents to contain emotions, rather than talk about them. These difficulties were perceived to manifested as anger and behavioural problems and compounded by a perceived lack of positive role models, stability, and support at home. These findings provide valuable insights into how children with these intersecting characteristics are perceived to experience a USB SEL intervention. It is also possible that these perceptions were influenced by a sensitivity towards externalising difficulties among boys, as discussed, or by discourses relating to, for example, "at-risk boys" [168].

Particularly pertinent was that the multiple risk factors experienced by deprived boys were perceived as a set of cir-cumstances external to school, and subsequently unchangeable by a teacher-led intervention. This is concerning, since there is a risk that these beliefs may make teachers less supportive of program implementation with children from certain groups if they believe that their difficulties and skills are inborne or unchangeable [169]. Indeed, our analyses suggest that

such perceptions *did* impact implementation, reducing intervention fidelity, since many teachers did not facilitate the family component of PATHS for these reasons.

## Lack of benefit for children with higher levels of difficulties

Our analyses suggest that PATHS was not perceived as beneficial for pupils with higher levels of difficulties. Although this is at odds with many previous studies of PATHS, which generally report differential gains among children with higher levels of social and/or emotional difficulties [170,171], the lack of intervention effects for children with high levels of challenging behaviours has also been reported in another trial [172]. In that trial, the authors concluded that PATHS, as a universal preventive intervention, was of insufficient intensity to address the multi-faceted needs of children with high levels of difficulties. Our analyses point to a similar conclusion, but also offer insight into teachers' beliefs about *why* such difficulties could not be addressed by PATHS. In our study, teachers reported that the difficulties experienced by these children were too severe, or too embedded, for a teacher-delivered universal intervention to remedy. When exploring the reasoning behind these beliefs, teachers pointed to interactions between multiple levels of children's ecology, specifically, interactions between gender, social and emotional competency, home life, and SES.

The subgroup of children identified by teachers as having higher levels of difficulties tended to be children from lower-SES/ deprived backgrounds, who were described as having increased need for social and emotional support. This reflects the current trend in the population of increased prevalence of socioemotional behavioural problems among children from disadvantaged and low-income households [43,49,51,52,70], with low socio-economic status strongly associated with both externalising [48,49] and internalising problems [39,50]. Living in socio-economic deprivation was perceived to be the cause of children's social and emotional difficulties, yet was also seen as the reason why children could not benefit from PATHS, largely because of parents/parenting. Similar views were echoed in Honess and Hunter's study of teacher perspectives on the implementation of PATHS in a different area of the UK [116]. In that study, teachers described children from "deprived" backgrounds as lacking in opportunities to develop social and emotional functioning at home. In our study, teachers attributed the difficulties experienced by deprived children predominantly to factors associated with their parents, including parental emotion socialisation and stress from home (see *Theme 4* in results section). This supports research that suggests the impact of poverty on children's mental health is transmitted through parents and parenting, with parental stress playing a significant role [44–46]. These children's inability to benefit from PATHS was attributed to the same parent-related factors: parental socialisation was believed to counter any SEL at school and parents were considered a more salient influence on children's social and emotional development.

## Perceived influence of parents and home life

Our analyses showed a tendency among teachers to speculate and attribute children's difficulties to their home lives. Assumptions were made about children's backgrounds that appeared to be in line with "troubled families" discourses [173] and negative social class stereotypes, for example, that parents do not value education, are linguistically deficient [174] and have "more feelings" [175]. Similar negative value judgments of social class were expressed in Hunter and Honess's study with pupils from lower-income families described as having "bad social backgrounds" [116]. In our study, connections were made between parents' social class, and their children's behaviour, and teachers expressed beliefs about social and emotional abilities that were bound up with notions of social class. For instance, empathy was viewed as characteristic of the middle classes, despite research demonstrating lower-class individuals typically score more highly on measures of empathy [176]. Although it is possible that these views were formed from teachers' personal experiences with their pupils, research has shown that that teachers' expectations for pupils can be biased by stereotypes and assumptions formed by misunderstandings of identities and backgrounds [177].

Whilst teachers clearly wanted to help pupils, good intentions can also form and reinforce unhelpful stereotypes. For instance, it has been suggested that messages conveyed by policy initiatives targeting low-income pupils, or discourses in teacher training relating to underachievement or behaviour among certain demographic groups may contribute to teachers' biases and low expectations [105]. The Pupil Premium strategy designed to support disadvantaged/ low-income pupils in the UK propagates the message that such children have difficulties with behaviour, wellbeing, and mental health, and have more safeguarding issues [177]. This is not to say that children from deprived or lower SES homes in the study did not experience social and emotional difficulties, or that poverty or parenting does not influence social and emotional development. Rather, it is the tendency to attribute children's difficulties on parents, parenting or home life that is problematic, since it could mean that classroom- or school-related factors are overlooked [178], or that pupils are unable to change.

Additionally, if negative stereotypes were communicated to pupils, this may negatively impact their sense of school belonging. School belonging, the extent to which pupils feel personally accepted, respected, included, and supported by others in the school social environment [179], is important for school-based outcomes as well as pupils' wellbeing [180]. Lower school belonging is related to both internalising and externalising behaviours [181] and pupils from lower SES backgrounds [179], and those with SEND [182] report lower school belonging than their peers. Furthermore, pupils' relationships with teachers are a key element of school belonging [183], so teachers should ensure that negative attitudes are not communicated to pupils.

### Home-school partnerships and SEL

Our findings highlight the need to engage parents in SEL and foster collaborative partnerships between school and home to better understand the influence of children's home life on social and emotional development during the primary school years. From an ecological perspective, the relationship between home and school is a significant force within a child's mesosystem [184]. Mutual support for the reinforcement of social and emotional skills by teachers and families is important for children's development [185]. SEL interventions with a family component have been found to be more efficacious in increasing social competence and decreasing challenging behaviour of preschool children than those without a family component [186], yet for primary school children, data on parent or family components is sparse. In their review of 252 USB SEL interventions, Cipriano and colleagues found that 40.8% of interventions reported no family component, and of those that did include a family component, only 5% of these involved engaging families with materials at home, whilst 6.9% offered training to families [3]. Furthermore, in a trial of Second Step, focus groups and interviews with parents and school staff suggest that school staff underestimate parents' interest in knowing what their children are learning in SEL and perceive barriers to communication with parents. Parents reported lacking knowledge about SEL, despite valuing that knowledge and wishing to support their children's SEL development [187]. In light of our findings that teachers emphasised the influence of parental- and home-related factors on intervention impact, our analyses suggest that increasing family involvement in school-based SEL is an area in need of development.

### Cultural relevancy for lower SES pupils

Another important finding relating to differential intervention effectiveness for children from lower SES backgrounds is that PATHS was perceived to lack cultural relevancy. The PATHS stories were seen as promoting a white middle-class suburban perspective, without consideration of the experiences of those from less privileged or culturally diverse backgrounds. Subsequently, PATHS was perceived to lack relevance for many pupils, and this was perceived to negatively impact engagement. It was left to teachers to make adaptations to the stories in order to increase engagement. This finding adds further support for calls to improve the cultural relevancy of SEL for marginalised groups, for instance, by using the lived experiences and frames of reference of students [188] in order to provide opportunities for students from all backgrounds to benefit [3,189].

## Less or no benefit for children with SEND/ low attainment

Our analyses indicate that teachers believed children with SEND, low academic attainment, and/or low literacy levels ben-efitted less from PATHS than others, or not at all. This was because the content and materials were perceived to be too difficult to understand, or required a high level of attentional control. Indeed, cognitive elements in USB SEL interventions that involve a high proportion of cognitive content, such as literacy, or learning skills, such as attention, have been found to negatively impact on social emotional skills and behaviours and reduce overall intervention effectiveness [190]. The lack of accessibility has been reported in other qualitative studies of SEL interventions, in which teachers describe having to differentiate learning to meet student needs [118].

Our findings in relation to Pupils with SEND may seem surprising, since the PATHS curriculum was originally developed as a targeted intervention for children in special education (specifically, those with hearing impairment [191]), before being adapted for other special education settings. Earlier trials of PATHS report the program to be effective for children with SEND. For example, Greenberg and colleagues reported improvements in emotional vocabulary and fluency in discuss-ing emotional experiences, efficacy beliefs regarding the management of emotions, and developmental understanding of some aspects of emotion, among their sample of children identified as having learning disabilities, mild mental retardation, severe behaviour disorders, or multi-handicaps (170, p.121). Additionally, a trial of children in special education classes reported significant reductions in teacher reports of externalising and internalising problems and substantial reductions in self-reported depression in a similar sample of children with learning disabilities, mild mental retardation, emotional and behavioural disorders, physical disabilities/health impairments and multiple handicap [192].

However, the findings of these previous two studies are from quantitative trials and therefore not directly comparable to our results. Aswell as differences in dosage (in both Kam and Greenberg and colleagues' studies, PATHS was delivered 2–3 times a week for 20–30 minutes, with a total of 60 sessions, whilst the pupils in our study received a lower dosage of 40 sessions delivered at most twice a week), there were differences in the program delivered (in Kam and colleagues' trial, teachers used a modified version of PATHS differentiated for special education and additional interventions were implemented alongside PATHS). There are other marked contextual differences between these studies and our study in terms of implementation. Firstly, in the two previous studies, PATHS was delivered to special education pupils by their special education teachers in self-contained classrooms. In our study, PATHS was delivered by mainstream teachers to Pupils with SEND in their mixed-attainment classes, alongside pupils with no SEND. Whilst there is a lack of evidence on the impact of variants of inclusion versus special education on outcomes for pupils with SEND [193], there is some evidence to suggest that students with disabilities whose teacher is certified in special education have greater achieve-ment than similar students whose teacher is not special education certified [194]. It is therefore possible that having a special education teacher deliver PATHS had a positive impact on pupil outcomes in the previous trials. Secondly, pupils with SEND in English schools may be incomparable to those identified as 'special education pupils' in the USA. Meanings of terms such as 'special educational needs', 'disability', and 'learning disability' differ between the USA and England, and the term 'learning difficulty', whilst widely used in England, tends not to be used in the USA [193]. In our study, teachers described how responsiveness and engagement varied by type of SEND, so it could be that particular elements of the PATHS curriculum rather than the entire program, or particular pedagogical approaches, were less effective for certain types of SEND in the context of an English mainstream classroom (see *Theme 2* in results section). Investigation into how pupils with SEND experience USB SEL interventions at a more granular level would be valuable here.

The context of implementation in our study is reflective of a typical English primary school and, thus, demonstrates how PATHS is perceived to work in a real-world context, outside of the country of development and independent of developer involvement. It is important to identify these contextual factors, since they may influence the adoption, implementation and sustainability of interventions when they go to scale [195], particularly given the UK's commitment to inclusive edu-cation and the removal of barriers to learning and participation in mainstream education [196]. Thus, it is crucial that SEL

interventions are accessible for all pupils in the setting in which they are implemented. Interventions that are heavily text based or rely on a secure level of literacy or attentional control reinforce an ableist agenda and create barriers to participation. Furthermore, given the potential for USB SEL interventions to improve academic achievement [5], interventions designed with an ableist orientation may serve to increase the achievement gap and contribute to further educational inequity [197]. Indeed, our analyses also indicate that more able children were able to accumulate further advantage through using learning from PATHS to increase academic attainment (see *Theme 1* in results section).

Notwithstanding, whilst our analysis clearly indicates that the accessibility of PATHS content and materials negatively impacted on the engagement and responsiveness of pupils with SEND, it is possible that teachers could have underestimated the positive impacts of PATHS among those pupils. Other SEL intervention studies have reported that teachers' reports of changes in social-emotional competences of students with SEND are less sensitive than pupil self-report and assessment [198].

### Potential influence of teachers' expectations

Teachers appeared to hold expectations of intervention effectiveness for the groups of pupils they talked about, and these expectations were low for pupils with SEND and pupils from socio-economically deprived backgrounds. For students with SEND, SEL was often seen as irrelevant, reflecting a broader tendency to view these students as separate from the mainstream and having low learning potential across the curriculum. This mirrors other studies reporting beliefs among teachers that school-wide USB SEL programs are not meaningful to students with disabilities [199] and aligns more broadly with research showing lower academic expectations for students labelled with learning difficulties [200–202]. It is possible these low expectations for social and emotional competency were communicated to these students, potentially hindering their development through self-fulfilling prophecies. Pupils from marginalised groups, including those from lower SES backgrounds, have been found to be more sensitive to expectation effects [98,103].

Teachers may have also inadvertently created self-fulfilling effects through their adjustments to the intervention content, omitting more challenging material for lower-attaining students. However, while such adaptations could limit learning, they might also increase engagement; adapting SEL interventions for diverse learners is complex, as modifications can both undermine and enhance effectiveness. How USB SEL interventions can be adapted to meet the needs of diverse learners without undermining the effectiveness of the core components has been subject to debate within the field: whilst in some cases, adaptations might undermine intervention effectiveness, in other cases they may improve outcomes [111]. Adaptations that are responsive to participant needs can increase responsiveness, and may even be necessary if elements of the intervention are confusing [84].

Another possibility is that teachers' initial low expectations for certain students' social and emotional competency created self-maintaining effects, influencing their perceptions and reports of the intervention's impact, regardless of actual outcomes. Research on expectation effects primarily focuses on academic performance, highlighting a need to investigate whether these effects similarly impact social and emotional development. If they do, teachers' low expectations of social and emotional competency may hinder pupils' social and emotional development.

### Influence of implementation quality

Teachers' affective engagement, sensitivity, and responsiveness to pupils are important components of high quality SEL implementation [89], during and outside of SEL lessons [86]. This suggests that good teacher-pupil relationships will enhance implementation quality. Our analyses indicate reciprocal benefits; teachers reported that having a good relationship with pupils meant they knew more about them and could help link the PATHS stories to their lives, increasing relevancy, and subsequently engagement and responsiveness among pupils, whilst also providing opportunities to get to know pupils better.

However, for pupils with high levels of difficulties, this was described as more challenging, teachers expressed frustration, and had low expectations of benefit for them. It is easy to see how teachers' negative beliefs or lower expectations could negatively impact the teacher-pupil relationship. Our results further add to the evidence base on implementation quality by suggesting that less-observable factors, such as teachers' beliefs about *pupils* may also influence implementation: teacher's implicit biases in relation to socio-demographic characteristics, and their expectations of social and emotional competency for groups of pupils, may affect the quality of implementation, through the communication to pupils of differential expectations of social and emotional competency, or through negatively impacting teacher-pupil relationships. It could be that these children needed a more therapeutic style of delivery; meta-analytic evidence has shown that school-based interventions for depression and anxiety are more effective when delivered by clinicians compared to teachers, such that clinician-delivered programs have significant, positive impacts, while teacher-delivered programs have null effects [203].

**Implications**

These findings can help inform decisions about the organisation of SEL support within a school, particularly in relation to consistency in approaches between different forms of support and interventions, as well as which staff are best placed as implementers. There are policy implications for investing in and supporting the integration of mental health services within schools to address the needs of students with more severe social, emotional, and behavioural difficulties.

That teachers reported PATHS as difficult to access for Pupils with SEND has important implications for policy and practice. At a school and local authority level, it is important that SEL intervention selection and implementation align with the UK's commitment to inclusive education and the removal of barriers to learning and participation for all students, including those with SEND. It is hoped that SEL intervention developers will gain useful knowledge to improve the accessibility and inclusivity of program content and materials to enable all students to have the opportunity benefit from SEL. This will involve considerations of different cultural and linguistic backgrounds, as well as considering neurodiversity and students with differing levels of literacy and academic attainment.

Because PATHS appears to require a baseline of social-emotional skills, schools and teachers should consider strategies for identifying students who may need pre-intervention support or alternative approaches. This may involve the development of more effective social-emotional assessment methods to help teachers identify children with internalising difficulties. This also highlights the importance of funding and developing early intervention programs that build foundational social-emotional skills before interventions like PATHS are implemented.

The findings also have implications for decisions surrounding communication and partnerships between schools and parents/caregivers when implementing SEL. Increased efforts to involve parents in SEL may help diffuse teachers' negative stereotypes and facilitate a more coordinated approach to supporting students' social and emotional development.

This study also highlights the importance of examining the influence of teacher factors, namely teachers' beliefs about pupils, on outcomes in USB SEL studies. School staff are the gatekeepers of such interventions; it is they who ultimately decide who is offered the opportunity for meaningful engagement, and who may be excluded. Prior to implementation, school staff should be provided with training and professional development to address potential biases and promote culturally responsive teaching practices, since evidence suggests that when teachers are made aware of their implicit biases, they may be more likely to address them [159]. Indeed, lack of training in SEL has been identified by teachers themselves as a barrier to SEL implementation [204] and is associated with both treatment fidelity and resulting student outcomes in USB SEL interventions [203]. Teachers should also prioritise building strong, positive relationships with all students, especially those who may be perceived as less responsive or having greater challenges, as this is important for effective SEL implementation.

Finally, our results have implications for those researching the differential impacts of USB SEL interventions. Researchers should be attuned to their own biases, with caution to the "adoption of particular perspective from which some things

become salient and others merge into the background" ([205], p.1); rather than searching for subgroup effects on the basis of single characteristics such as gender or SES, an intersectional approach will undoubtedly prove more useful. Furthermore, the interaction of multiple characteristics may be heavily dependent on the school or demographic context. It is important that investigator's observations and anticipations of effects do not take precedence over the lived experiences of the children and young people receiving interventions [206]. There is a need to use different methods to identify 'invisible' groups. For instance, future SEL trials would benefit greatly from utilising qualitative methods to explore the perspectives of children and young people who receive the interventions. This would provide deeper insights into the perceived benefits for different children, and would triangulate existent knowledge.

### Recommendations for schools and educators

#### 1. Provide more effective training before SEL implementation

Teachers should receive comprehensive training and professional development in school *before* implementing SEL interventions. This training should explicitly address potential implicit biases and promote culturally responsive teaching methods to ensure equitable opportunities for all students. Teachers should receive guidance on how to make adaptations to intervention content and materials that increase accessibility for pupils with diverse cultural, linguistic, and neurodevelopmental backgrounds, without compromising the core components of the program. Training should also emphasise the preventative nature of SEL, so staff understand the relevancy of SEL for all pupils.

#### 2. Develop strong teacher-pupil relationships

Educators should prioritise building positive and supportive relationships with all students, particularly those who may be perceived as less engaged or facing greater challenges. Such strong connections are fundamental for effective SEL implementation. Educators should actively examine their beliefs and expectations regarding students based on gender, socioeconomic status, and SEND status to avoid the creation of self-fulfilling or self-sustaining prophecies. Teachers should be mindful of how expectations are communicated through interactions, eye contact, feedback, and learning opportunities and recognise that all students, regardless of background or perceived difficulties, are capable of growth in their social and emotional skills.

#### 3. Ensure consistency across support systems

Schools should promote a consistent approach to social and emotional support across different interventions and staff within school. That includes timetable considerations to allow pupils to engage fully with support.

#### 4. Strengthen home-school partnerships in SEL

Schools and educators should actively involve parents and caregivers in the implementation of SEL programs. Information about SEL and the programs implemented should be communicated with parents and caregivers, since parents may lack knowledge about SEL but are often interested in supporting their children's development in this area. While acknowledging the impact of home environment, school staff should avoid making negative assumptions about parents or attributing all difficulties solely to home factors. Developing collaborative partnerships with families will help gain a better understanding of students' home lives to reinforce social and emotional skills across settings.

### Strengths and limitations of the present study

The present study adds to limited literature that explores teachers' perceptions of the impact of USB SEL interventions and adds a number of new insights into factors influencing implementation at the child-level and teacher-level. Previous research has found that children's individual and socio-demographic characteristics can influence intervention impact. The current study adds further nuances through insights into *how* these characteristics are perceived to influence impact, and how they may interact with teachers' implementation.

Our study was conducted using robust and reproducible methodology. It demonstrates how thematic framework analysis can be utilised in SEL intervention research, which often seeks to answer contextual and evaluative questions relating to the effectiveness of systems and programs [143]. Framework analysis enabled us to systematically apply the framework to manage the large dataset in a meaningful, rigorous and transparent way. The matrix output provides an audit trail of links between the data and interpretations, enhancing the dependability of the findings [148,149].

There are, however, several limitations to our study. The principal author (SH) was not involved in data collection, so analysis may not have fully captured participants' meanings; a transcription is only partially representative of an interaction between participant and researcher, capturing something, but not everything, and altering that 'something' [207]. Although the data were generated through a large qualitative sample (N = 105), the present study is based on teachers from Greater Manchester, which may limit the applicability of findings to broader educational contexts. However, the findings have the potential to be generalisable in other ways. The framework method allowed for a larger sample size than is usual in qualitative research (sample sizes are typically under 50 interviews [208]); this may enhance the representativeness and transferability of findings to similar schools within similar contexts. Indeed, similar views were echoed in another evaluation of PATHS in a different region [116], as well as in other studies of SEL programs in the US [118].

Whilst our study aimed to investigate teachers' perceptions of differential impacts, as key stakeholders delivering interventions on the ground, we relied solely on their reports of the intervention's impact on children, with no observational data. Information about implementation quality and child engagement gathered solely from teachers may lack reliability [86]. Direct observation allows for a more detailed and reliable assessment of children's and teachers' behaviours within the context they occur [209]. What teachers reported was limited to observable behaviours displayed by children in the school setting and thus may not provide a whole picture.

Additionally, the sample is predominantly female, which may affect perceptions of SEL effectiveness, since female teachers, compared with male teachers, have been found to demonstrate stronger beliefs in and greater intention to advocate for the implementation of SEL [210]. In qualitative inquiry, though, it is important that samples represent the phenomenon rather than the general population [211] and in most OECD countries, between 75 and 85% of teachers are female [212]. This demographic bias may, therefore, be more representative of the teaching workforce. Further research is needed to explore how perceptions may differ across a more diverse workforce.

Another limitation is that data from the main trial indicate on average teachers delivered PATHS once per week in sessions of 35–45 minutes, a lower dosage than recommended by the developers (two 30 minute sessions per week, plus weekly supplementary activities). Dosage reportedly reduced towards the end of the trial, from 65% of lessons being delivered in the first year, to 39% of lessons in the second year [125]. Thus, findings relating to the effectiveness of the PATHS curriculum must be interpreted with caution. However, this does not detract from the important insights gained in respect of teachers' views of how a USB SEL intervention is experienced by different groups of children in a real-world context. Furthermore, most studies do not report implementation data and of those that do, sub-optimal delivery is the norm rather than the exception [88], so it could be argued that our results have greater external validity.

Whilst the findings provide valuable insights into subgroups of pupils that may experience more, less, or no benefit from a USB SEL intervention, there may be other unidentified subgroups, such as children from ethnic minority groups, or those with internalising symptoms. Within the subgroups identified, it is likely that there are discrepancies between children's and teachers' reports [213]. Whilst teacher reports provide valuable insights into pupils' functioning in school, and may be more reliable for certain aspects of children's social and emotional functioning, such as self-awareness [214], data from multiple informants would offer triangulation [213]. Furthermore, through introspection, the child has access to the most detailed information about themselves than of any of the possible respondents [214]. Future studies should explore children's perspectives as receivers of interventions. The current study relied on data on the perspectives of teachers, but future work will want to integrate multiple perspectives.

A further limitation of our study is that it is not possible to know whether teachers' biases or differential expectations were communicated to, or perceived by pupils, or if they impacted on implementation beyond teachers' delivery of the parent resources. In order to serve as a self-fulfilling prophecy, expectations must be perceived by pupils [103], and not all teachers treat pupils differently according to their expectations or are influenced by biases to the same degree [215]. Future intervention research could explore pupils' perceptions of teachers' implementation as part of an investigation into the impact of implementation quality on intervention outcomes.

The study data were generated throughout 2013−14 and may not reflect current perceptions. The last decade has seen an increase in societal awareness and knowledge of mental health [216,217], even more so since the public discourses on the mental health impacts of the Covid-19 pandemic [218]. Within education specifically, policy reforms have emphasised schools' responsibilities to promote mental health and wellbeing, for *all* pupils, through supporting social and emotional development [219,220]. There has been even more emphasis on mental health in the curriculum for all pupils [221] and school staff, including teachers, are increasingly expected to attend to children's social-emotional wellbeing. We expect that teachers today may have a different approach to SEL, and may be more perceptive of less visible social and emotional difficulties, such as internalising difficulties, among children.

Additionally, absent from the data was any discussion of ethnicity. This is surprising, since over a quarter of the children (26.3%) taking part in the trial were from a non-Caucasian background [125]. Whilst it may seem encouraging that teachers endeavour to view children as individuals irrespective of ethnicity, this prevents exploration of whether the benefits of USB SEL interventions extend to different ethnic groups. It has been suggested that SEL programs promote white middle-class values and culture and may lack relevancy for children who have different cultural orientations and values [189], therefore, it is important that this is investigated. Ignoring racial/ethnic identities in favour of individualism, or "colour-blindness" [222] is equated with a denial of racism and of the systems of stratification that privilege some groups over others [223]. Given the increase in societal awareness of the structural inequalities that disproportionality affect ethnic minority groups, it is likely that teachers today may be more attuned to the differential experiences of children from ethnic minority groups.

## Conclusion

The overall aim of our study was to understand how teachers described and explained the varied outcomes of the PATHS curriculum on different groups of students, within their school contexts. Our results offer insight into some of the processes interacting with the implementation of a USB SEL intervention that may contribute to intervention impact for certain groups of pupils. Through the thematic framework analysis, we have found that teachers do think that there are subgroups of children that benefit differently, or not at all, from this type of intervention. According to teachers, these subgroups are defined by the intersection of multiple characteristics of a child's ecology, rather than single characteristics. These characteristics may also influence teachers' expectations of impact.

## Supporting information

**S1.  interview schedule.**
(DOCX)

**S2.  theoretical framework of child-level influences.**
(DOCX)

## Author contributions

**Conceptualization:** Suzanne Hamilton, Jan R. Boehnke, Pamela Qualter, Neil Humphrey.

**Data curation:** Suzanne Hamilton.

**Formal analysis:** Suzanne Hamilton.

**Funding acquisition:** Neil Humphrey.

**Methodology:** Suzanne Hamilton.

**Supervision:** Jan R. Boehnke, Neil Humphrey, Pamela Qualter.

**Writing – original draft:** Suzanne Hamilton.

**Writing – review & editing:** Jan R. Boehnke, Neil Humphrey, Pamela Qualter.

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
