## [Decision Letter · Decision Letter 0]

Dear Dr. Hamilton,

Thank you for submitting your manuscript to PLOS ONE. After careful consideration, we feel that it has merit but does not fully meet PLOS ONE’s publication criteria as it currently stands. Therefore, we invite you to submit a revised version of the manuscript that addresses the points raised during the review process.

We look forward to receiving your revised manuscript.

Kind regards,

Ramandeep Kaur

Academic Editor

PLOS ONE

4. We note that you have referenced (118.O’Brien A, Panayiotou M, Santos J, Humphrey N, Hamilton S. A systematic review exploring the relationship between implementation variability and outcomes in universal, school-based social and emotional learning interventions. manuscript **in preparation** ; Available from: https://www.crd.york.ac.uk/prospero/display_record.php?ID=CRD42023416661) which has currently not yet been accepted for publication. Please remove this from your References and amend this to state in the body of your manuscript: (ie “Bewick et al. [Unpublished]”) as detailed online in our guide for authors

Additional Editor Comments:

Dear Author,

Thank you for submitting the manuscript, Teachers’ perceptions of the differential impacts of a universal, school-based social and emotional learning intervention: a thematic framework analysis. This study presents a valuable exploration of teacher perspectives on the differential impact of SEL interventions on students. While the research is well-structured and methodologically sound, several areas require further refinement to enhance clarity, applicability, and overall contribution to the field. Below are specific suggestions for improvement:

Major Areas for Improvement:

1. Addressing Generalizability Concerns

The study is based on teachers from Greater Manchester, which limits the applicability of findings to broader educational contexts.

Please acknowledge this limitation explicitly in the discussion and provide insights on how findings might translate to other regions or diverse school settings.

If possible, consider incorporating comparisons with existing studies from other regions to highlight similarities and differences in SEL implementation.

2. Reducing Potential Bias from Self-Reported Data

Since the study relies entirely on teacher perspectives, the findings may reflect subjective biases rather than objective student outcomes.

Consider discussing how teacher-reported perceptions compare with actual student progress metrics (if available from existing literature).

If no student data are available, suggest how future research could incorporate student or parental perspectives for a more holistic understanding.

3. Participant Representation and Demographic Limitations

The sample is predominantly female (81%) and postgraduate-educated (51%), which may affect perceptions of SEL effectiveness.

It would be helpful to acknowledge and discuss how this demographic bias might influence the interpretation of results.

If possible, suggest how future studies could include a more diverse sample to explore potential differences in perspectives based on teacher background.

Clarity, Structure, and Formatting Improvements:

4. Improving Readability and Conciseness

Certain sections, particularly the literature review and discussion, contain long, dense paragraphs that could be streamlined for better readability.

Consider breaking down complex discussions into shorter, focused subsections with clear takeaways for the reader.

Some long sentences in the methodology and findings should be restructured for clarity.

5. Theoretical Overload in Discussion

While the discussion is thoroughly grounded in existing literature, there is an over-reliance on theoretical explanations, which may make it less accessible for practitioners.

Consider balancing theoretical insights with more practical implications, particularly in how teachers can apply the findings to real-world SEL implementation.

A dedicated subsection summarizing key recommendations for educators would enhance the manuscript’s usability.

Methodological and Data Considerations:

6. Clarifying the Lack of Data Accessibility

The manuscript states that no new data were created or analyzed, raising concerns about data availability for replication.

Please clarify whether the dataset used in this study is accessible for further research.

If the data are not publicly available, consider discussing how future studies could build on this work with additional datasets.

7. Strengthening the Discussion on SEL Implementation Challenges

While the study discusses teacher perceptions of SEL effectiveness, there is limited discussion on specific implementation barriers.

Expanding this section to cover real-world challenges (e.g., time constraints, teacher training gaps, resource limitations) would enhance the manuscript’s practical relevance.

Recommendation for Revision

The study offers valuable insights into how teachers perceive SEL interventions, but the generalizability, data bias, participant diversity, and readability concerns should be addressed.

Suggested Next Steps:

Explicitly acknowledge sample limitations and discuss implications for generalizability.

Address bias from self-reported teacher perspectives and suggest potential solutions.

Streamline dense sections and improve clarity of long sentences.

Balance theoretical analysis with more practical recommendations.

Clarify data accessibility and potential for replication.

Expand discussion on real-world implementation barriers for SEL programs.

Once these revisions are incorporated, the manuscript will make a stronger contribution to the field of educational psychology and SEL research.

Looking forward to the revised submission.

Reviewers' comments:

Reviewer's Responses to Questions

**Comments to the Author**

1. Is the manuscript technically sound, and do the data support the conclusions?

Reviewer #1: Partly

2. Has the statistical analysis been performed appropriately and rigorously?

Reviewer #1: Yes

3. Have the authors made all data underlying the findings in their manuscript fully available?

Reviewer #1: No

4. Is the manuscript presented in an intelligible fashion and written in standard English?

Reviewer #1: Yes

Reviewer #1: Need major revision; please follow manuscript writing procedure. It seems like thesis writing more. Focus on key findings that you want to share with readers. Concise your writing. Thanks for submitting.

**Do you want your identity to be public for this peer review?** For information about this choice, including consent withdrawal, please see our Privacy Policy

Reviewer #1: No

---

## [Author Response · Author response to Decision Letter 1]

27 Jun 2025

1.1 When submitting your revision, we need you to address these additional requirements.

The title page, headings, tables, and file names have been reformatted according to PLOS ONE style requirements.

1.2 We note that you have indicated that there are restrictions to data sharing for this study. PLOS only allows data to be available upon request if there are legal or ethical restrictions on sharing data publicly.

The data are stored on secure encrypted servers held by the University of Manchester. We are unable to make data available for anyone outside of the research team at the University of Manchester. This is because the data were collected prior to 2014, and consent was not obtained from participants or from University Research Ethics Committee for the public sharing of their data. Whilst the data are anonymised, they contain potentially identifiable and sensitive information relating to participants and their pupils. NH was a principal investigator of the main PATHS evaluation; we were granted secure remote access to this data by Dr Alexandra Hennessey (second principal investigator of the main evaluation and data custodian).

1.3 Please include your full ethics statement in the ‘Methods’ section of your manuscript file. In your statement, please include the full name of the IRB or ethics committee who approved or waived your study, as well as whether or not you obtained informed written or verbal consent. If consent was waived for your study, please include this information in your statement as well.

We have added to the manuscript:

The trial was approved by the University Research Ethics Committee at the University of Manchester (reference number 11470). Consent from participants was provided in writing (Lines 266-268, p.13).

1.4 We note that you have referenced (118.O’Brien A, Panayiotou M, Santos J, Humphrey N, Hamilton S. A systematic review exploring the relationship between implementation variability and outcomes in universal, school-based social and emotional learning interventions. manuscript in preparation. Please remove this from your References and amend this to state in the body of your manuscript: (ie “Bewick et al. [Unpublished]”) as detailed online in our guide for authors.

This paper has now been published. The reference has been changed as follows: O’Brien A, Panayiotou M, Santos J, Hamilton S, Humphrey N. A systematic review informing recommendations for assessing implementation variability in universal, school-based social and emotional learning interventions. Social and Emotional Learning: Research, Practice, and Policy. 2025 May;100112. (Reference 92, p. 80)

1.5 Please include captions for your Supporting Information files at the end of your manuscript, and update any in-text citations to match accordingly.

Captions and in-text citations of supporting information have been changed accordingly and added to end of MS:

S1 interview schedule.

S2 theoretical framework of child-level influences.

S3 participant consent form.

2.1 Addressing Generalizability Concerns

The study is based on teachers from Greater Manchester, which limits the applicability of findings to broader educational contexts.

Please acknowledge this limitation explicitly in the discussion and provide insights on how findings might translate to other regions or diverse school settings.

If possible, consider incorporating comparisons with existing studies from other regions to highlight similarities and differences in SEL implementation.

We have added the following:

"Similar views were echoed in Honess and Hunter’s study of teacher perspectives on the implementation of PATHS in a different area of the UK (120). In that study, teachers described children from “deprived” backgrounds as lacking in opportunities to develop social and emotional functioning at home." (Lines 1178-1181, p.60)

"Similar negative value judgments of social class were expressed in Hunter and Honess’s study with pupils from lower-income families described as having “bad social backgrounds (120)” (Lines 1194-6, p.60)

"However, the lack of accessibility has been reported in other qualitative studies of SEL interventions, in which teachers describe having to differentiate learning to meet student needs (Hunter et al., 2022).”

(Lines 1263-1265, p.63-4)

"..the present study is based on teachers from Greater Manchester, which may limit the applicability of findings to broader educational contexts. However, the findings have the potential to be generalisable in other ways. The framework method allowed for a larger sample size than is usual in qualitative research (sample sizes are typically under 50 interviews (214)); this may enhance the representativeness and transferability of findings to similar schools within similar contexts. Indeed, similar views were echoed in another evaluation of PATHS in a different region (120), as well as in other studies of SEL programs in the US (122).” (Lines 1479- 1486, p.73-4)

2.2 Reducing Potential Bias from Self-Reported Data

Since the study relies entirely on teacher perspectives, the findings may reflect subjective biases rather than objective student outcomes.

Consider discussing how teacher-reported perceptions compare with actual student progress metrics (if available from existing literature).

If no student data are available, suggest how future research could incorporate student or parental perspectives for a more holistic understanding.

We direct reviewers to the following paragraphs in the manuscript:

"Furthermore, the main trial in which the data was generated reported no significant differences between children in PATHS and usual provision schools in respect of teachers’ reports of their internalising symptoms, but did find a statistically significant improvement in children’s self-reports of psychological well-being (129). Thus, it could be speculated that teachers were less perceptive of the impact of PATHS on children’s internalising symptoms. However, the main trial found a significant increase in pro-social behaviour and a small reduction in emotional symptoms among children identified as high risk (those with elevated symptoms at baseline) in both child and teacher reports (163). This aligns with our finding that teachers reported benefit for children with some degree of difficulties, but suggests that teachers were, to some extent, able to identify internalising difficulties and improvements.” (Lines 1080 – 1090, pp.55-6).

"Whilst the findings provide valuable insights into subgroups of pupils that may experience more, less, or no benefit from a USB SEL intervention, there may be other unidentified subgroups, such as children from ethnic minority groups, or those with internalising symptoms. Within the subgroups identified, it is likely that there are discrepancies between children’s and teachers’ reports (216). Whilst teacher reports provide valuable insights into pupils’ functioning in school, and may be more reliable for certain aspects of children’s social and emotional functioning, such as self-awareness (217), data from multiple informants would offer triangulation (216). Furthermore, “through introspection, the child has access to the most detailed information about themselves than of any of the possible respondents” (217), so gaining their perspectives, as receivers of interventions, through qualitative inquiry would be invaluable.” (Lines 1513-1524, p.75)

We have added the following:

"Future studies should explore children’s perspectives as receivers of interventions. The current study relied on data on the perspectives of teachers, but future work will want to ....” (Lines 1521-1524, p.75)

2.3 Participant Representation and Demographic Limitations

The sample is predominantly female (81%) and postgraduate-educated (51%), which may affect perceptions of SEL effectiveness.

It would be helpful to acknowledge and discuss how this demographic bias might influence the interpretation of results.

If possible, suggest how future studies could include a more diverse sample to explore potential differences in perspectives based on teacher background.

We have added the following to the manuscript:

"Additionally, the sample is predominantly female, which may affect perceptions of SEL effectiveness, since female teachers, compared with male teachers, have been found to demonstrate stronger beliefs in and greater intention to advocate for the implementation of SEL (216). In qualitative inquiry, though, it is important that samples represent the phenomenon rather than the general population (217) and in most OECD countries, between 75 and 85% of teachers are female (218). This demographic bias may, therefore, be more representative of the teaching workforce. Further research is needed to explore how perceptions may differ across a more diverse workforce.” (Lines 1495-1502, p. 74)

2.4 Clarity, Structure, and Formatting Improvements:

Improving Readability and Conciseness

Certain sections, particularly the literature review and discussion, contain long, dense paragraphs that could be streamlined for better readability.

Consider breaking down complex discussions into shorter, focused subsections with clear takeaways for the reader.

Some long sentences in the methodology and findings should be restructured for clarity.

We have edited the manuscript and broken down/ condensed sentences where possible. We have streamlined sections of the introduction, discussion and methodology to improve conciseness and clarity. The following sections in particular have been edited:

• The section of the introduction about Free school meal (FSM) eligibility (Lines 120-133, p.6)

• The entire section of the discussion entitled Potential influence of teachers’ expectations (Lines 1325-1354, pp.66-68)

2.5 Theoretical Overload in Discussion

While the discussion is thoroughly grounded in existing literature, there is an over-reliance on theoretical explanations, which may make it less accessible for practitioners. Consider balancing theoretical insights with more practical implications, particularly in how teachers can apply the findings to real-world SEL implementation. A dedicated subsection summarizing key recommendations for educators would enhance the manuscript’s usability.

We have added the following section:

"Recommendations for schools and educators

1. Provide more effective training before SEL implementation

Teachers should receive comprehensive training and professional development in school before implementing SEL interventions. This training should explicitly address potential implicit biases and promote culturally responsive teaching methods to ensure equitable opportunities for all students. Teachers should receive guidance on how to make adaptations to intervention content and materials that increase accessibility for pupils with diverse cultural, linguistic, and neurodevelopmental backgrounds, without compromising the core components of the program. Training should also emphasise the preventative nature of SEL, so staff understand the relevancy of SEL for all pupils.

2. Develop strong teacher-pupil relationships

Educators should prioritise building positive and supportive relationships with all students, particularly those who may be perceived as less engaged or facing greater challenges. Such strong connections are fundamental for effective SEL implementation. Educators should actively examine their beliefs and expectations regarding students based on gender, socioeconomic status, and SEND status to avoid the creation of self-fulfilling or self-sustaining prophecies. Teachers should be mindful of how expectations are communicated through interactions, eye contact, feedback, and learning opportunities and recognise that all students, regardless of background or perceived difficulties, are capable of growth in their social and emotional skills.

3. Ensure consistency across support systems

Schools should promote a consistent approach to social and emotional support across different interventions and staff within school. That includes timetable considerations to allow pupils to engage fully with support.

4. Strengthen home-school partnerships in SEL

Schools and educators should actively involve parents and caregivers in the implementation of SEL programs. Information about SEL and the programs implemented should be communicated with parents and caregivers, since parents may lack knowledge about SEL but are often interested in supporting their children's development in this area. While acknowledging the impact of home environment, school staff should avoid making negative assumptions about parents or attributing all difficulties solely to home factors. Developing collaborative partnerships with families will help gain a better understanding of students’ home lives to reinforce social and emotional skills across settings.

Lines 1428 – 1460, pp.71-3

2.6 Methodological and Data Considerations:

6. Clarifying the Lack of Data Accessibility

The manuscript states that no new data were created or analyzed, raising concerns about data availability for replication.

c The authors are unable to make data available for anyone outside of the research team at the University of Manchester. The data were collected prior to 2014, and consent was not obtained from participants or from University Research Ethics Committee for the public sharing of their data. Whilst the data are anonymised, they contain potentially identifiable and sensitive information relating to participants and their pupils, and, therefore, cannot be shared.

2.7 Strengthening the Discussion on SEL Implementation Challenges

While the study discusses teacher perceptions of SEL effectiveness, there is limited discussion on specific implementation barriers.

Expanding this section to cover real-world challenges (e.g., time constraints, teacher training gaps, resource limitations) would enhance the manuscript’s practical relevance.

We have added the following:

Additionally, if negative stereotypes were communicated to pupils, this may negatively impact their sense of school belonging. School belonging, the extent to which pupils feel personally accepted, respected, included, and supported by others in the school social environment (186), is important for school-based outcomes as well as pupils’ wellbeing (187). Lower school belonging is related to both internalising and externalising behaviours (188). Pupils from lower SES backgrounds (189), and those with SEND (190) report lower school belonging than their peers. Furthermore, pupils’ relationships with teachers are a key element of school belonging (191), so teachers should ensure that negative attitudes are not communicated to pupils.

These findings can help inform decisions about the organisation of SEL support within a school, particularly in relation to consistency in approaches between different forms of support and interventions, as well as which staff are best placed as implementers. There are policy implications for investing in and supporting the integration of mental health services within schools to address the needs of students with more severe social, emotional, and behavioural difficulties.

That teachers reported PATHS as difficult to access for Pupils with SEND has important implications for policy and practice. At a school and local authority level, it is important that SEL intervention selection and implementation align with the UK's commitment to inclusive education and the removal of barriers to learning and participation for all students, including those with SEND... Because PATHS appears to require a baseline of social-emotional skills, schools and teachers should consider strategies for identifying students who may need pre-intervention support or alternative approaches. This may involve the development of more effective social-emotional assessment methods to help teachers identify children with internalising difficulties. This also highlights the importance of funding and developing early intervention programs that build foundational social-emotional skills before interventions like PATHS are implemented.

The finding

---

## [Editor Report · Decision Letter 1]

Teachers’ perceptions of the differential impacts of a universal, school-based social and emotional learning intervention: a thematic framework analysis.

PONE-D-25-10776R1

Dear Dr. Hamilton,

We’re pleased to inform you that your manuscript has been judged scientifically suitable for publication and will be formally accepted for publication once it meets all outstanding technical requirements.

Kind regards,

Ramandeep Kaur

Academic Editor

PLOS ONE

Additional Editor Comments (optional):

Thank you for your detailed and thoughtful revisions to your manuscript titled “Teachers’ perceptions of the differential impacts of a universal, school-based social and emotional learning intervention: a thematic framework analysis.”

You have addressed all editorial and reviewer comments thoroughly. Your expanded discussion on generalizability, implementation challenges, and data limitations has added depth and nuance to the manuscript. The newly added “Recommendations for Schools and Educators” section greatly enhances the paper’s relevance to practitioners, aligning well with the translational aims of PLOS ONE.

We are satisfied that the manuscript now meets our editorial requirements for publication, and we are pleased to accept it for publication in PLOS ONE.

Congratulations on your work.

Reviewers' comments:

Based on the thorough and well-documented revisions, the manuscript now meets **PLOS ONE’s editorial and publication criteria** , including ethical transparency, data policy justification, scientific rigor, and clarity of reporting.

---

## [Editor Report · Acceptance letter]

PONE-D-25-10776R1

PLOS ONE

Dear Dr. Hamilton,

I'm pleased to inform you that your manuscript has been deemed suitable for publication in PLOS ONE. Congratulations! Your manuscript is now being handed over to our production team.

Kind regards,

on behalf of

Dr. Ramandeep Kaur

Academic Editor

PLOS ONE